# Wintertime enhancements of sea salt aerosol in polar regions consistent with a sea-ice source from blowing snow

Jiayue Huang[1] and Lyatt Jaeglé[1]

[1]Department of Atmospheric Sciences, University of Washington, Seattle, Washington, USA

*Correspondence to*: Lyatt Jaeglé (jaegle@uw.edu)

**Abstract.** Sea salt aerosols (SSA) are generated via air bubbles bursting at the ocean surface, as well as by wind mobilization of saline snow and frost flowers over sea-ice covered areas. The relative magnitude of these sources remains poorly constrained over polar regions, affecting our ability to predict their impact on halogen chemistry, cloud formation and climate. We implement a blowing snow and a frost flower emission scheme in the GEOS-Chem global chemical transport
model, which we validate against multi-year (2001-2008) in situ observations of SSA mass concentrations at three sites in the Arctic, two sites in coastal Antarctica, as well as during a cruise in the Arctic (ICEALOT, 2008). A simulation including only open ocean emissions underestimates SSA mass concentrations by factors of 2-10 during winter-spring for all ground-based and ship-based observations. When blowing snow emissions are added, the model is able to reproduce observed wintertime SSA concentrations, with the model bias decreasing from a range of −80% to −34% for the open ocean
simulation to −2% to +9% for the simulation with blowing snow emissions. We find that the frost flower parameterization cannot fully explain the high wintertime concentrations and displays a seasonal cycle decreasing too rapidly in early spring. Furthermore, the high day-to-day variability of observed SSA is better reproduced by the blowing snow parameterization. Over the Arctic (>60ºN) (Antarctic, >60ºS), we calculate that submicron SSA emissions from blowing snow account for 1.0 Tg yr$^{-1}$ (2.5 Tg yr$^{-1}$), while frost flower emissions lead to 0.21 Tg yr$^{-1}$ (0.25 Tg yr$^{-1}$) compared to 0.78 Tg yr$^{-1}$ (1.0 Tg yr$^{-1}$)
from the open ocean. Blowing snow emissions are largest in regions where persistent strong winds occur over sea ice (East of Greenland, over the central Arctic, Beaufort Sea, as well as the Ross and Weddell Seas). In contrast, frost flower emissions are largest where cold air temperatures and open leads are co-located (over the Canadian Arctic Archipelago, coastal regions of Siberia, and off the Ross and Ronne ice shelves). Overall, in situ observations of mass concentrations of SSA suggest that blowing snow is likely to be the dominant SSA source during winter, with frost flowers playing a much
smaller role.

## 1 Introduction

Breaking waves over the open ocean are recognized as the main mechanism for the global production of sea salt aerosol (SSA) (Lewis and Schwartz, 2004; de Leeuw et al., 2011 and references therein). Observations of SSA in polar regions, however, exhibit several characteristics that are not consistent with this canonical open ocean source. Indeed, submicron or

total SSA mass concentrations at Arctic (Sirois, et al., 1999; Quinn et al., 2002) and Antarctic sites (Wagenbach et al., 1998; Weller et al., 2008; Jourdain et al., 2008; Udisti et al., 2012) often exhibit a maximum during local winter, when polar ocean waters are mostly covered by sea-ice. Furthermore, the ionic composition of SSA observed at polar sites during winter shows a systematic depletion of the sulfate to sodium mass ratio relative to bulk sea water (Wagenbach et al., 1998; Rankin et al.,

2000; Jourdain et al., 2008; Hara et al., 2012; Jacobi et al., 2012; Seguin et al., 2014). Finally, Arctic and Antarctic ice core records display factors of 2.5-4 increase in SSA deposition fluxes during glacial periods relative to warmer interglacial period (Wolff et al. 2006; Fischer et al., 2007; Abram et al., 2013).

To explain these seasonal and glacial-interglacial variations, frost flowers has been proposed as a new source of SSA (Wagenbach et al., 1998; Rankin et al. 2000, 2002; Wolff et al., 2003; Shaw et al., 2010). They are highly saline ice crystals

that can rapidly form on freshly freezing sea ice (Martin et al., 1995; Domine et al., 2005; Roscoe et al., 2011). Frost flowers wick up brine from the sea ice and can be lofted in the atmosphere by surface winds to become SSA (Rankin et al. 2000; Domine et al., 2004; Xu et al. 2013 and references therein). The seasonality of frost flower formation and their sulfate to sodium ratios are similar to those of observed SSA in polar regions (Rankin et al., 2002; Rankin and Wolff, 2003; Wolff et al., 2003; Alvarez-Aviles et al., 2008; Beaudon and Moore, 2009; Seguin et al., 2014). Field observations have cast some

doubt on the role of frost flowers as SSA, noting that frost flowers are rigid and difficult to break (Domine et al., 2005; Alvarez-Avilez et al., 2008). In particular, Obbard et al. (2009) observed no mechanical breakage of frost flowers in winds up to 6 m s[-1] over the Hudson Bay. Furthermore, laboratory experiments performed by Roscoe et al. (2011) demonstrated that no aerosol were produced when frost flowers were exposed to winds speeds up to 12 m s[-1]. Their result is consistent with electron microscope imaging by Yang et al. (2017), which show that evaporating frost flowers form a cohesive chunk of salt

that is unlikely to be a source of SSA.

Another hypothesis is that blowing snow, often observed over sea ice-covered regions (Nishimura and Nemoto, 2005; Savelyev et al., 2006), could act as a direct source of SSA (Simpson et al., 2007a; Yang et al., 2008). The snow over sea ice becomes salty by upward migration of brine from the sea ice to the snow surface, incorporation of frost flowers, as well as SSA deposition from the adjacent open ocean (Domine et al., 2004). The first two of these mechanisms lead to depletion of

the sulfate to sodium ratio relative to bulk sea water as mirabilite ($Na_2SO_4 \cdot 10H_2O$) precipitates from brine at temperatures below -8°C during sea ice and frost flower formation (Alvarez-Aviles et al., 2008). Once lifted by wind, these salty snow particles can produce SSA via sublimation (Yang et al., 2008).

Many questions remain on the formation, composition, occurrence, and mobility of frost flowers and salty blowing snow. Most studies examining these two sources have focused on their potential role as an indirect source of gas-phase bromine

resulting in ozone depletion events during late winter and early spring, with conflicting results as to which source would be most important. Kaleschke et al. (2004) developed a one-dimensional thermodynamic model to calculate frost flower coverage. They found that more than 66% of forward trajectories from areas with high frost flower coverage intercepted

regions with enhanced BrO over the Arctic and Antarctic during polar sunrise. Similarly, Jones et al. (2006) showed that ozone depletion events observed at Halley station in Antarctica were associated with air masses having recent contact with newly forming sea ice. However, using backtrajectories to examine the origin of enhanced BrO abundance measured at Barrow, Alaska, Simpson et al. (2007a) found that saline snow and ice on first-year sea ice was a more likely source of bromine than frost flowers. Yang et al. (2010) implemented a blowing snow bromine source in the p-TOMCAT chemistry-transport model. With this blowing snow source they were able to successfully simulate bromine explosion events retrieved from the Global Ozone Monitoring Experiment (GOME).

To our knowledge, the only modeling studies examining blowing snow and frost flowers as direct sources of SSA are the work of Xu et al. (2013, 2016), Levine et al. (2014), and Legrand et al. (2016). Xu et al. (2013) developed an empirical frost flower formulation in the WRF-Chem model. They found that adding frost flower emissions doubled the surface concentrations of $Na^+$ at Barrow, in better agreement with observations. However, their study was limited to two days during winter 2009. Their work was expanded in Xu et al. (2016), where the same frost flower emission scheme was implemented in the Community Earth System Model (CESM) for the year 2000 and compared to SSA observations at Barrow and Alert. They found that the frost flower simulation lead to improved agreement Barrow, but overestimated observations at Alert by 150%. Levine et al. (2014) found that the local winter peak in $Na^+$ mass concentrations at polar sites was attributable to blowing snow, but they were not able to constrain the relative sources of open ocean and blowing snow because rates of SSA wet deposition were tuned regionally in the p-TOMCAT model. Using the p-TOMCAT model, Legrand et al. (2016) report that 50-70% of wintertime $Na^+$ concentrations at two East Antarctic sites were due to sea ice emissions from blowing snow. However, the model overestimated observed $Na^+$ concentrations by factors of 2-3 and was not able to reproduce the observed seasonal cycle.

In this study, we implement a blowing snow and a frost flower parameterization in the GEOS-Chem global chemical transport model. We evaluate the ability of these two sources to reproduce multi-year (2001-2008) in situ measurements of $Na^+$ mass concentrations at three Arctic sites (Barrow, Alaska; Alert, Canada; Zeppelin, Svalbard), two coastal Antarctic sites (Neumayer and Dumont d'Urville), as well as $Na^+$ measurements obtained during the International Chemistry Experiment in the Arctic Lower Troposphere (ICEALOT) cruise during spring 2008. We then examine the relative contributions of open ocean, blowing snow and frost flower sources to the distribution of SSA over polar regions.

## 2 Model simulations and observations

### 2.1 The GEOS-Chem chemical transport model

We use the GEOS-Chem global 3-D chemical transport model (Bey et al., 2001) driven by the Modern Era Retrospective-Analysis for Research and Applications (MERRA; Rienecker et al., 2011) meteorological fields. The MERRA fields have a

native horizontal resolution of 1/2º latitude by 2/3º longitude with 72 vertical levels. We regrid these fields to a 2º×2.5º horizontal resolution and vertical levels above 80 hPa are merged to retain 47 vertical levels in total for computational expediency. The temporal resolution of MERRA data is 3 hours except for surface variables and mixing depths, which have a 1-hour resolution. The sea ice concentration boundary conditions in MERRA are derived from the weekly product of Reynolds et al. (2002), which is based on Special Sensor Microwave Imager (SSMI) instruments on Defense Meteorological Satellite Program (DMSP) satellites. The weekly products are linearly interpolated in time to each model time step. In this study we use GEOS-Chem v10-01 (http://www.geos-chem.org).

The GEOS-Chem open ocean SSA simulation is described in Jaeglé et al. (2011). Ocean emissions are based on the wind speed-dependent source function of Gong (2003) and Monahan et al. (1986), with an empirical dependence on sea surface temperature (SST) derived by Jaeglé et al. (2011) via comparisons to open ocean cruise observations of coarse mode SSA mass concentrations. This SST dependence leads to a decrease in SSA emissions of a factor of 2.6 as SST decreases from 25ºC to 5ºC, consistent with the factors of 2-3 measured in laboratory experiments for particles with a radius greater than 0.5 μm (Bowyer, 1984, 1990; Woolf et al., 1987; Mårtensson et al., 2003; Sellegri et al., 2006). Most of the in situ observations of SSA mass concentrations used to derive the polynomial SST dependence (Fig. 6 in Jaeglé et al., 2011: $f(SST)=0.3 + 0.1 \times SST - 0.0076 \times SST^2 + 0.00021 \times SST^3$) were for SST>5ºC. In this work, we find that the SST dependence results in a factor of 2 overestimate of summertime SSA observations at coastal polar sites with SST ranging from -2ºC to 5ºC. This indicates that the suppression at cold SST might not be strong enough. We thus modify the expression derived in Jaeglé et al. (2011) to impose $f(SST)=0.25$ for SST<5ºC (see Fig. S1 in the Supplement). This is consistent with the laboratory study of Mårtensson et al. (2003), who report a 50-60% decrease in aerosol production (for r>0.1 μm) when seawater temperature decreased from 5ºC to -2ºC. We note that another potential explanation for the summertime overestimate in SSA mass concentrations is inefficient wet removal from low-intensity summer precipitation in GEOS-Chem (Croft et al., 2016).

Within GEOS-Chem, we assume that open ocean emissions occur only in gridboxes covered by more than 50% water. We thus neglect emissions from bubble bursting in leads within sea ice. This is based on observations that the small fetch of leads results in SSA production which is an order of magnitude lower compared to open ocean (Nilsson et al. 2001). Including SSA emissions over leads results in a 1% increase in SSA emissions over polar regions (>60º). Even if we were to assume that leads were as efficient as the open ocean in producing SSA, this would only result in a 10% increase in SSA emissions.

Dry deposition of SSA over land accounting for particles growth under high humidity conditions follows the size-segregated scheme described in Zhang et al. (2001). The dry deposition velocity over the ocean is calculated based on the Slinn and Slinn (1980) deposition model for natural waters. Over snow and ice surfaces, Fisher et al. (2011) implemented a dry deposition velocity of 0.03 cm s$^{-1}$ based on the measurements of Nilsson and Rannik (2001). The wet deposition scheme includes convective updraft scavenging, rainout and washout from precipitation (Liu et al., 2001), as well as snow

scavenging (Wang et al., 2011). For this work, we track SSA mass in two size bins: accumulation mode ($r_{dry}$ = 0.01−0.5 μm) and coarse mode ($r_{dry}$ = 0.5–4 μm), except in the comparison to in situ mass concentrations of SSA for which we use $r_{dry}$ = 0.01−0.3 μm and $r_{dry}$ = 0.3−3 μm (see section 2.3). In the rest of the manuscript we will refer to the accumulation and coarse mode SSA as submicron and supermicron SSA based on their diameters.

The GEOS-Chem SSA simulation was evaluated by Jaeglé et al. (2011) against in situ measurements of SSA from 6 open ocean cruises (mean normalized bias of +33%) and 15 ground-based stations (mean normalized bias of -5%) as well as aerosol optical depth (AOD) from MODIS and AERONET. Detailed comparisons of GEOS-Chem black carbon and organic aerosol (Wang et al., 2011) as well as sulfate and ammonium aerosol (Fisher et al., 2011) to ground-based and aircraft observations over the Arctic during winter and spring suggest that transport and removal processes are reasonably captured
by the model.

**2.2 Implementation of blowing snow and frost flower parameterizations in GEOS-Chem**

We implement SSA emissions from blowing snow following the parameterization of Yang et al. (2008, 2010), with a few modifications. The SSA production from blowing snow is a function of relative humidity, temperature, age of snow, snow salinity and wind speed. The wind needs to be strong enough (> about 7 m s$^{-1}$) to saltate and suspend snow particles from the
sea ice surface. The size distribution of suspended blowing snow particles follows a two-parameter gamma distribution (Yang et al., 2008 and references therein). We set a uniform surface snow salinity of 0.1 psu (practical salinity unit) over Arctic sea ice based on observations of surface snow salinity (Mundy et al., 2005; Krnavek et al., 2012). This is an order of magnitude lower than the salinity assumed in Yang et al. (2008, 2010). Their salinity was based on bulk snow measurements (Massom et al., 2001), which overestimate surface snow salinity because of the rapid decreases of salinity with height above
the ice surface. For example, Mundy et al. (2005) found that springtime snowpack over first year sea ice in the central Canadian Arctic displayed a salinity of 7.6±3.3 psu in the bottom snow layers and decreased to 0.26±0.37 psu and 0.11±0.25 psu in the middle and surface snow layers. This profile is consistent with a salinity source from the underlying sea ice and little influence from atmospheric deposition. Toom-Sauntry and Barrie (2002) find that freshly fallen snow itself tends to have low salinity (<0.01 psu). For simplicity, we assume the same salinity for surface snow on first-year and multi-year sea
ice, although we recognize that in reality the surface snow may be less salty on multi-year sea ice due to less efficient upward transport of brine. Indeed, Krnavek et al. (2012) reported that the ion concentrations of surface snow sampled in the Alaskan Arctic display large variability depending on sea ice type: 0.01 psu for snow on multiyear sea ice, 0.1 psu for snow on thick first year sea ice and 0.8 psu for snow on thin first year sea ice. Snowpack on Antarctic sea ice is thicker than over Arctic sea ice. Indeed, Antarctica is surrounded by the Southern Ocean which brings moisture, while the Arctic is
surrounded by land with wintertime precipitation that is 3 times lower than over Antarctic sea ice (Huffman et al., 2001). As the salinity of snow decreases with snowpack thickness, we assume that the salinity of snow on Antarctic sea ice is 0.03 psu, a factor of 3 lower than over the Arctic following Yang et al. (2010).

Based on the time between precipitation events in the MERRA fields, we estimate a mean snow age of 3 days for the Arctic and 1.5 days for the Antarctic. In comparison, Yang et al. (2008) used a globally uniform snow age of 3 days, while Levine et al. (2014) assumed 5 days. Our younger snow age over Antarctic sea ice (1.5 days) increases the ease of lifting snow particles and SSA production increases by 40% compared to a 3 day snow age. We also assume that 5 SSA particles are produced per snowflake. This fractionation can occur when snowflakes are broken by strong winds and abraded into smaller particles with round corners (Mellor, 1965), or when SSA particles experience cracking during sublimation under low relative humidity as observed in the laboratory experiments of Wise et al. (2012). The number of particles produced per snowflake (N) does not change the total SSA production substantially, but it influences the size distribution of SSA particles produced from these lofted snow particles. We choose the value of N=5 based on wintertime observations of supermicron and submicron SSA at Barrow (see Fig. S2 in the Supplement). This leads to a doubling of submicron SSA production compared to the assumption of N=1 in Yang et al. (2008). Our simulation yields blowing snow SSA emissions with 38% of SSA mass in the submicron range (0.01−0.5 μm) and 62% in the supermicron range (0.01−0.5 μm) for the Arctic. As we assume a lower salinity in the Antarctic, more of the blowing snow emissions are in the submicron range (60%) in that region. Overall, our modifications to the Yang et al. (2008, 2010) parameterization lead to SSA emissions from blowing snow that are an order of magnitude lower than in Yang et al. (2010).

We implement a frost flower SSA source in GEOS-Chem following the parameterization of Xu et al. (2013, 2016), which is based on the potential frost flower (PFF) coverage derived by Kaleschke et al. (2004). Frost flowers are formed on very young sea ice once ambient air temperatures are cold enough (<–20ºC). Below that threshold, PFF increases rapidly as temperature decreases. We set a limit of 10 cm for the thickness of newly formed sea ice beyond which we assume that frost flower crystals can no longer be formed. Based on the thermodynamic model of Kaleschke et al. (2004), it takes 1-2 days for sea ice to reach a thickness of 10 cm for air temperatures of –40 to –20ºC. Following Xu et al. (2013), we assume that SSA from frost flowers have a lognormal size distribution with a geometric mean diameter of 0.015 μm and a geometric standard deviation of 1.9, hence all frost flower emissions occur in the submicron range ($r_{dry} \leq 0.5$ μm). We use the same scaling factor of $5 \times 10^6$ $m^2$ $s^{-1}$ as Xu et al. (2013) for our frost flower source. We also conducted a sensitivity simulation (Fig. S3), in which we assumed that frost flowers can only form under mild wind speed conditions (<5 m $s^{-1}$), as strong winds inhibit frost flower formation and bury existing frost flowers with snow (Perovich and Richeter-Menge, 1994; Rankin et al., 2000).

We conduct three simulations. Our standard simulation (STD) includes only the open ocean source of SSA. In a second simulation (STD-SNOW), we add the blowing snow source to the STD simulation. A third simulation (STD-FF) adds frost flower emissions to the STD simulation.

**2.3 In situ observations**

We use in situ observations of $Na^+$ mass concentrations from 5 polar sites: Barrow, Alaska (71.3ºN, 156.6ºW; 11m a.s.l.; Quinn et al., 2002); Alert, Nunavut, Canada (82.5ºN, 62.5ºW; 210m a.s.l., WMO/GAW, 2003); Zeppelin Mountain, Svalbard, Norway (78.9ºN, 11.9ºE; 475m a.s.l, WMO/GAW, 2003); Neumayer (70.7°S, 8.3°W; 42m a.s.l; Weller et al., 2008) and Dumont d'Urville (66.7ºS, 140ºE; 43m a.s.l.; Legrand et al., 2012). These observations are available for 2001-2008 (except for Neumayer station, 2001-2007). At Barrow $Na^+$ mass concentrations are available for both submicron and supermicron aerosol, while all the other sites measure total mass concentrations. The $Na^+$ mass concentrations are determined by ion chromatography with uncertainties of 5%-11% (0.01 μg/m$^3$ in absolute uncertainty). The aerosol sampling frequency ranges from daily (Zeppelin, Dumont d'Urville, submicron at Barrow) to weekly (Alert, Neumayer, Barrow supermicron). In winter months, the coastlines near these sites are mostly covered by sea ice.

We also use the submicron $Na^+$ mass concentrations measured aboard the R/V *Knorr* during the International Chemistry Experiment in the Arctic LOwer Troposphere (ICEALOT) cruise in March-April 2008 (http://saga.pmel.noaa.gov/Field/icealot). The research cruise took place over the North Atlantic Ocean and the ice-free Arctic Ocean (41º-81ºN).

For comparison between the GEOS-Chem model and the observations, we convert observed $Na^+$ mass concentrations to SSA mass concentrations using a factor of 3.256 based on the mass ratio of $Na^+$ in seawater (Riley and Chester, 1971). For frost flowers, this ratio is 3.237 (Rankin et al., 2000). Krnavek et al. (2012) find a ratio of 3.24-3.278 for snow on first year sea ice. As this ratio varies by less than 0.5% for these different SSA sources, we use a constant factor of 3.256.

The reported aerodynamic cut-off diameters of the measurements are 1 μm and 10 μm at Barrow and during ICEALOT (Quinn et al., 2002), and 7-10 μm at the other sites (Wagenbach et al., 1998; WMO/GAW, 2003; Weller et al., 2008). In order to compare to model simulations, we need to convert these aerodynamic diameters to dry geometric radii. This conversion depends on aerosol density, relative humidity during sampling, and whether the particle is spherical (Seinfeld and Pandis, 2006). For example, a 10 μm aerodynamic diameter could correspond to a dry geometric radius of 2.3 μm (80% RH, 1.2 g/cm$^3$ pure NaCl solution, a factor of 2 growth between dry and 80% RH), 3 μm (dry cubical NaCl particle, 2.2 g/cm$^3$, Lewis and Schwartz, 2004), or 3.8 μm (30% RH, ammonium sulfate and sea salt aerosol, 1.7 g/cm$^3$, Quinn et al., 1996). Thus for comparison to observations we choose the mid-range estimate and conduct a simulation with two size bins: $r_{dry} = 0.01-0.3$ μm and $r_{dry} = 0.3-3$ μm.

**3 Model evaluation with ground-based and ship-based in situ observations**

Observations at the three Arctic sites display enhanced SSA mass concentrations of 1-3 μg m$^{-3}$ during the cold part of the year from November to April (Fig. 1a-c). In contrast, when the sea ice retreats during summer and late fall, SSA

concentrations are much lower (<0.5 µg m$^{-3}$ at Alert and Zeppelin). This seasonality is opposite to what is expected from an open-ocean source. Indeed, we find that the STD simulation fails to capture the high wintertime concentrations at all three Arctic sites but reproduces the summer/late fall observations reasonably well. During winter at Barrow and Alert, the STD simulation predicts very low SSA concentrations (< 0.1 µg m$^{-3}$), while at Zeppelin, which is closer to the open Atlantic

Ocean, STD mass concentrations reach 0.5-1 µg m$^{-3}$.

At Neumayer station (Fig. 1d), SSA observations show a broad maximum of 1-1.5 µg m$^{-3}$ during the cold months (March-September). The seasonality is opposite at Dumont d'Urville (Fig. 1e), with a summertime maximum of 2.9 µg m$^{-3}$, as it is exposed to a longer open ocean season compared to Neumayer (Wagenbach et al. 1998). Between March and November, SSA concentrations remain fairly constant around 0.8 µg m$^{-3}$. The summertime maximum at Dumont d'Urville is captured

by the STD simulation, confirming the open ocean source. However, the STD model predicts SSA concentrations <0.3 µg m$^{-3}$ during cold months, factors of 3-10 lower than observations at both Antarctic sites.

The addition of a blowing snow source in GEOS-Chem (STD-SNOW) results in improved agreement with observations. The normalized mean bias (NMB=$(\overline{Model}/\overline{Obs} - 1)\times100$) decreases significantly at all five sites: Barrow (STD: -64%, STD-SNOW:+9%), Alert (STD: -85%; STD-SNOW: +25%); Zeppelin (STD: -34%; STD-SNOW: +12%); Neumayer (STD: -

63%; STD-SNOW: -2%); Dumont D'Urville (STD: -40%; STD-SNOW: +12%). The STD-SNOW simulation captures the observed SSA seasonal cycle reasonably well, with modeled wintertime SSA mass concentrations increasing to 1-2 µg m$^{-3}$.

Overall, the frost flower simulation (STD-FF) displays a large geographical variability, with little influence at Dumont d'Urville and Zeppelin, but much larger influence at Barrow, Neumayer and especially at Alert, where modeled SSA concentrations reach 1 µg m$^{-3}$. Indeed, the STD-FF simulation predicts very large SSA emissions over the Canadian Arctic

Archipelago (see section 4 for a detailed discussion and maps of spatial distributions). The NMB in the STD-FF simulation ranges from -49% (Neumayer) to -27% (Zeppelin), not displaying as large an improvement as the STD-SNOW simulation. Furthermore, the seasonal cycle of frost flower SSA concentrations decreases too rapidly during early spring compared to observations at Alert, Barrow and Neumayer.

We examine in more detail the daily variability in submicron SSA at Barrow for January-July 2001 (Fig. 2a). Between

January and late April, the observations show large day-to-day variations with concentrations ranging from <0.5 µg m$^{-3}$ to 2-4 µg m$^{-3}$. These SSA enhancements last for 1-7 days. We find that the timing and magnitude of these events are often reproduced by the blowing snow simulation (observations: 0.98±0.9 µg m$^{-3}$; STD-SNOW: 0.94±0.8 µg m$^{-3}$) and are driven in part by variability in windspeed (Fig. 2b, gray shaded areas). Some events are not associated with high local winds (29 January - 5 February; 24-28 March), and are due to transport from nearby regions. For example, the high levels of SSA

submicron concentrations seen in the blowing snow simulation on 29 January - 5 February are associated with wind-blown snow coming from sea ice in the northern Beaufort Sea. In contrast, the frost flower simulation fails to capture the variability

and magnitude of observed SSA events (STD-FF: 0.28±0.29 μg m$^{-3}$). An examination of weekly SSA mass concentrations at Alert and daily concentrations at Dumont d'Urville for 2001 yields similar conclusions (see Fig. S4 in the Supplement).

Figure 3a shows submicron SSA mass concentrations measured aboard the R/V *Knorr* during the ICEALOT experiment in March-April 2008. The first part of the cruise took place over the North Atlantic, where the largest enhancements in SSA mass concentrations (1-2 μg m$^{-3}$ on 25-26 March and 29 March) were due to open ocean SSA emissions and are reproduced by the STD simulation. As the R/V *Knorr* travelled towards the Norwegian, Barents and Greenland Seas (3-20 April), the STD simulation predicts very low SSA concentrations (<0.2 μg m$^{-3}$) and can no longer reproduce the observed concentrations (0.5-2 μg m$^{-3}$). The STD-FF simulation produces enhancements that are too weak, but the STD-SNOW simulation captures some of these enhancements, in particular on 6-7 April (the R/V *Knorr* was along the Norwegian coast) and 15-19 April (near the coast of Svalbard). During both periods, Gilman et al. (2010) report concurrent decreases in observed O$_3$ and in the acetylene to benzene ratio, indicative of destruction of surface O$_3$ by Br and oxidation of acetylene by both Br and Cl. Figure 3c shows that based on our STD-SNOW simulation, a major blowing snow event developed on 15-19 April over the central Arctic, poleward of 80ºN. At that time the R/V *Knorr* was positioned within a few kilometres off the sea ice edge and the observed O$_3$ decreased from 43 ppbv to 1.5 ppbv (Gilman et al., 2010). The STD-SNOW simulation predicts an increase in SSA concentrations of up to 1.5 μg m$^{-3}$ (Fig. 3a, shaded gray area) reproducing the timing and magnitude of the observed enhancement.

Overall, we find that the blowing snow source can explain the large wintertime enhancements in observed SSA mass concentrations over both the Arctic and Antarctic regions. Furthermore, the STD-SNOW simulation captures the episodic nature of the observed enhancements. The frost flower source reproduces some of the observed enhancements over the Arctic, but is not able to match the high SSA concentrations over coastal Antarctica, and does not have a strong enough day-to-day variability. It is possible that both blowing snow and frost flower emissions act together. However, when we add the contributions from both sources, we find that modelled SSA mass concentrations are a factor of 2-3 too high compared to observations at Barrow and Alert (Fig. S5). In particular, the frost flower simulation leads to a peak in SSA in February at Barrow, which is not observed. Our simulations thus suggest that the dominant influence is from blowing snow.

## 4 Emissions and distributions of SSA over polar regions

Table 1 summarizes the annual SSA budgets over the Arctic and Antarctic as calculated in GEOS-Chem for the year 2005 poleward of 60º latitude (see Table S1 in the Supplement for the global budgets). We find that annual SSA emissions vary by 10-30% for 2004-2008, but the overall seasonality and spatial distribution of emissions are similar from year to year. The total (0.01-4 μm) blowing snow source is 2.6 Tg yr$^{-1}$ for the Arctic and 4.2 Tg yr$^{-1}$ for Antarctica. We find that the larger blowing snow source over Antarctica, despite the lower snow salinity, is a result of faster winds over Antarctic sea ice. Furthermore, the younger age of snow assumed over Antarctic sea ice (1.5 days compared to 3 days over Arctic sea ice)

contributes to 30% of the difference in blowing snow emissions between the Arctic and Antarctic. The frost flower emissions are slightly stronger over the Antarctic (0.25 Tg yr$^{-1}$) than the Arctic (0.21 Tg yr$^{-1}$) due to strong katabatic winds over the Antarctic. The open ocean accounts for 30 Tg yr$^{-1}$ over the Arctic and 40 Tg yr$^{-1}$ over Antarctica. Examining submicron SSA ($r_{dry}$ = 0.01−0.5 μm), we see that this is the size range where blowing snow (Arctic: 1.0 Tg yr$^{-1}$; Antarctic: 2.5 Tg yr$^{-1}$) and frost flower (0.21 Tg yr$^{-1}$; 0.25 Tg yr$^{-1}$) emissions have their largest impact relative to the open ocean (0.78 Tg yr$^{-1}$; 1.0 Tg yr$^{-1}$). This difference in size distributions is related to the different physical mechanisms for SSA emissions from open ocean emissions (breaking waves and bubble bursting) compared to blowing snow (saltation of fallen snow and suspension) or frost flowers (saltation of broken frost flower crystals and suspension). The original crystalline form of snow particles/frost flower fragments are expected to be shattered by repeated impacts with the ground and other particles during saltation. Sublimation of ice from these particles leads to relatively small SSA compared to bubble bursting in the open ocean. In the following sections we focus on the seasonality and spatial distribution of submicron SSA emissions (Fig. 4 and 5).

## 4.1 Arctic

Figure 4a shows the seasonal evolution of our three SSA sources over the Arctic (>60ºN). SSA emissions from the open ocean maximize in September-October as a result of strong winds combined with minimum sea ice extent. During winter months, SSA emissions from the open ocean are largest over the ice-free North Atlantic Ocean, extending towards the Barents Sea (Fig. 4b). SSA emissions from blowing snow reach their maximum in December-April (Fig. 4a) with the largest emissions occurring over sea ice covered regions with the strongest winds (Fig. 4c): East of Greenland, over the central Arctic and the Beaufort Sea. The modeled blowing snow SSA surface mass concentrations reach 2-3.5 μg m$^{-3}$ over these regions (Fig. 5c). We find that atmospheric transport leads to inland incursions of blowing snow SSA over northern Canada, Alaska, and Siberia (Fig. 5c).

Frost flower emissions maximize in December-March and are 2-4 times smaller than blowing snow emissions during these months (Fig. 4a). We find that frost flower emissions are highly localized with the strongest emissions over the Canadian Arctic Archipelago (Fig. 4d), where surface concentrations of SSA reach 2-3 μg m$^{-3}$ (Fig. 5d), explaining the large influence we noted at Alert and Barrow (Fig. 1). Weaker emissions occur over coastal Siberia and in leads located within the central Artic sea ice. In our simulation, the location of frost flower emissions largely depends on the simultaneous occurrence of very cold air temperatures (<−20°C) and open leads. Other regions in the Arctic have cold temperatures during winter, however they are mostly covered by sea ice with limited open leads areas. Our emissions from frost flowers over the Arctic (0.21 Tg yr$^{-1}$) are consistent the accumulation model emissions reported by Xu et al. (2016) (0.24 Tg for November-February, their Table 2). We also find a similar geographic distribution.

We note that the mean lifetime of both blowing snow and frost flower submicron SSA is 6-7 days in the Arctic, nearly twice as long as open ocean SSA (Table 1). Open ocean SSA form over lower latitude warmer regions, while sea ice SSA emissions occur at higher latitudes under much colder conditions, with less efficient removal processes in mixed-phased and ice clouds. The current parameterization in GEOS-Chem assumes that in-cloud scavenging of SSA does not occur in cold

clouds (T<258 K) (Wang et al., 2011), thus wintertime sea-ice generated SSA are only removed by below-cloud scavenging (which is slow for accumulation mode aerosols) and dry deposition. Recent laboratory studies have shown that SSA could act as ice nuclei by deposition freezing (Wise et al., 2012) and immersion freezing (DeMott et al., 2016), and might thus undergo in-cloud scavenging in mixed and ice clouds. This process is not currently included in GEOS-Chem.

## 4.2 Antarctic

In the southern hemisphere polar regions, open ocean SSA emissions display a weak seasonal cycle due to persistent strong winds over the Southern Ocean (Fig. 4e). During austral winter, emissions from the open ocean are strongest at ~50ºS leading to modeled surface SSA concentrations of 1-3 µg m$^{-3}$ (Fig. 4f and 5f). Blowing snow emissions maximize in June-October (Fig. 4a) and are strongest over the sea ice of the Ross and Amundsen Seas, because of the strong katabatic winds flowing off the Antarctic Plateau, as well as strong winds in the Indian Ocean sector (Fig. 4g). In these regions, modeled

submicron SSA concentrations from salty snow reach 1-3 µg m$^{-3}$, explaining the increase of 1-2 µg m$^{-3}$ seen at Neumayer and Dumont d'Urville (Fig. 4e and Fig. 1d-e). The model predicts that frost flower emissions are concentrated near the Ross, Ronne and Amery ice shelves and along coastlines (Fig. 4h), accounting for 1-2 µg m$^{-3}$ surface submicron SSA over these regions (Fig. 5h). Neumayer thus receives influence from frost flowers formed off the Ronne ice shelf (Fig. 5h and 1d), while Dumont d'Urville has a weaker influence from frost flowers forming along the local coastline (<0.1 µg m$^{-3}$).

Spatially, we find that the locations of blowing snow and frost flower emissions are complementary to each other due to the different requirements of sea state (sea ice compared to open leads).

## 5 Discussion and conclusions

In this work, we implement two new SSA emission schemes in the GEOS-Chem chemical transport model: a blowing snow parameterization following the work of Yang et al. (2008, 2010) and a frost flower parameterization based on Xu et al.

(2013) and Kaleschke et al. (2004). We find that the GEOS-Chem simulation with open ocean emissions fails to capture the elevated SSA mass concentrations observed at five coastal stations in the Arctic and Antarctic during winter (2001-2008) and during the ICEALOT research cruise in March-April 2008. When blowing snow emissions are added, the model is able to reproduce the wintertime observed SSA levels as well as their large day-to-day variability driven by wind speed. We find that the frost flower parameterization cannot fully explain the high wintertime concentrations and displays a seasonal cycle

decreasing too rapidly in early spring. Furthermore, our frost flower simulation cannot reproduce the large daily variability of observed SSA.

Over the Arctic, we estimate that annual blowing snow emissions of submicron SSA are 1.0 Tg yr$^{-1}$, compared to 0.8 Tg yr$^{-1}$ from the open ocean. Over the Antarctic, these emissions are 2.5 Tg yr$^{-1}$ for blowing snow and 1.0 Tg yr$^{-1}$ for the open ocean. Blowing snow emissions are mostly controlled by wind speed and are thus larger over the Antarctic due to the strong katabatic winds off the Antarctic Plateau and strong westerlies over the Southern Ocean. Frost flower SSA emissions are 0.21 Tg yr$^{-1}$ over the Arctic (0.25 Tg yr$^{-1}$ for the Antarctic) and depend on the co-location of cold air temperatures and open leads.

The parameterizations for blowing snow and frost flowers have several intrinsic assumptions, such as the salinity of snow and the scaling factor for frost flowers, which will affect the relative magnitudes of these two sources in polar regions. The geographic distribution, seasonal cycle, and daily variability of these sources, however, are controlled by sea ice extent and meteorological parameters (winds and temperature). In this study, we showed that the temporal and geographical variability of SSA observations at five polar sites is more consistent with blowing snow than with frost flowers. Based on this comparison, we conclude that blowing snow is likely to be the dominant source of SSA in polar winter, although frost flowers cannot be entirely ruled out. In particular, they may contribute indirectly to SSA emissions by salinating wind-blown snow (Obbard et al., 2009).

These polar sources of SSA are subject to substantial uncertainties due to the limited observations available. One key uncertainty in our simulations is snow salinity. Indeed, SSA emissions from blowing snow have a near-linear dependence on the salinity of snow. Thus a doubling of the assumed salinity would lead to a doubling in SSA emissions from blowing snow. Furthermore, we assume a uniform salinity of snow over both first-year and multi-year sea ice. This likely overestimates the contribution of blowing snow SSA over the western Arctic, which is dominated by multi-year sea ice. More extensive observations of surface snow salinity at multiple locations over both first-year and multi-year sea ice can help further refine these assumptions. Sampling of SSA size distributions during blowing snow events can help determine the number of particles per snowflake, which we determined empirically in this study. This number will not affect to total SSA emissions, but will change the relative importance of submicron and supermicron SSA emissions. There is insufficient knowledge on frost flower occurrence, growth and their mobilization by winds. In particular, the role of favourable wind conditions, as well as the ice thickness for frost flower to grow, are highly uncertain, and thus the predicted locations of frost flower emissions in our simulation are also uncertain.

Reducing these remaining uncertainties would help constrain how sea ice emissions of SSA affect the chemistry of the polar atmosphere by acting as a source of halogens, leading to ozone and mercury depletion events (Barrie et al., 1988; Fan and Jacob, 1992; Simpson et al., 2007b; Schroeder et al., 1998; Steffen et al., 2008). Improved process-based understanding of

these emissions would also lead to better constraints on the potential climatic impact of wintertime SSA on clouds, in particular mixed-phase and ice clouds, which have a strong influence on downward longwave radiative forcing. Indeed, recent studies have shown the role of SSA as ice nuclei (Wise et al., 2012; DeMott et al., 2016). Thus over the Arctic and Antarctic regions, where the abundance of other ice nuclei such as dust or black carbon are low, SSA from local sea ice

sources could influence the formation, radiative forcing, and precipitation of mixed-phase and ice clouds.

**Acknowledgements**

This work was supported by funding from the NASA Atmospheric Composition Modeling and Analysis Program under award NNX15AE32G. The authors wish to thank the NOAA Pacific Marine Environmental Laboratory (PMEL) Atmospheric chemistry group for providing the in situ aerosol observations at Barrow and during the ICEALOT field

campaign. We also thank Environment Canada for providing the in situ observations at Alert, the French observation service CESOA (http://www-lgge.obs.ujf-grenoble.fr/CESOA/spip.php?rubrique3) for the Dumont D'Urville observations, and the Norwegian Institute for Air Research (NILU) for the Zeppelin Mountain observations.  The authors would like to acknowledge useful discussions with Maurizio Di Pierro, Steve Warren, Cecilia Bitz, and Becky Alexander.

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

Table 1: Arctic (>60°N) and Antarctic (>60°S) SSA budgets for the open ocean, blowing snow, and frost flower sources for the year 2005.

| | Open ocean | | | Blowing Snow | | | Frost Flowers |
|---|---|---|---|---|---|---|---|
| | 0.01-0.5 μm | 0.5-4 μm | Total | 0.01-0.5 μm | 0.5-4 μm | Total | 0.01-0.5 μm |
| Arctic (>60°N) | | | | | | | |
| Emission (Tg yr⁻¹) | 0.78 | 29 | 30 | 1.0 | 1.6 | 2.6 | 0.21 |
| Dry deposition (Tg yr⁻¹) | 0.13 | 13 | 13 | 0.35 | 0.77 | 1.1 | 0.091 |
| Wet deposition (Tg yr⁻¹) | 1.2 | 20 | 21 | 0.43 | 0.89 | 1.3 | 0.050 |
| Lifetime (days) | 3.5 | 0.36 | 0.48 | 6.6 | 0.72 | 2.6 | 7.0 |
| Burden (Gg) | 12 | 32 | 45 | 14 | 3.3 | 17 | 2.7 |
| Surface concentration (μg m⁻³) | 0.19 | 1.0 | 1.2 | 0.40 | 0.17 | 0.57 | 0.11 |
| Antarctic (>60°S) | | | | | | | |
| Emission (Tg yr⁻¹) | 1.0 | 39 | 40 | 2.5 | 1.7 | 4.2 | 0.25 |
| Dry deposition (Tg yr⁻¹) | 0.30 | 24 | 24 | 0.41 | 0.62 | 1.0 | 0.082 |
| Wet deposition (Tg yr⁻¹) | 2.6 | 25 | 28 | 1.2 | 1.0 | 2.2 | 0.074 |
| Lifetime (days) | 3.5 | 0.38 | 0.55 | 4.4 | 0.52 | 2.4 | 6.3 |
| Burden (Gg) | 28 | 50 | 78 | 19 | 2.4 | 21 | 2.7 |
| Surface concentration (μg m⁻³) | 0.46 | 1.6 | 2.1 | 0.45 | 0.093 | 0.54 | 0.094 |

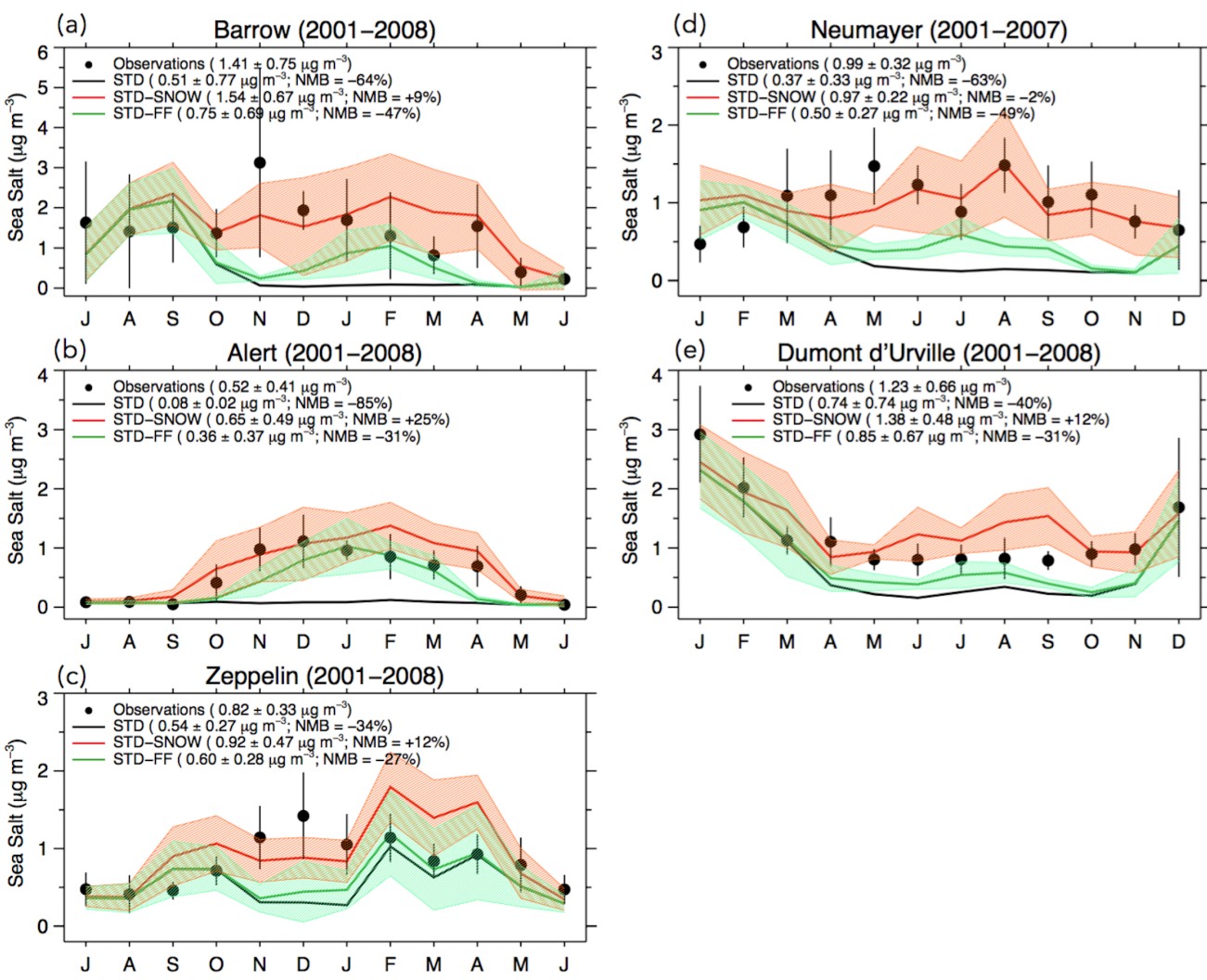

**Figure 1: Monthly mean mass concentrations of SSA at Arctic (a. Barrow, b. Alert, c. Zeppelin) and Antarctic sites (d. Dumont d'Urville, e. Neumayer). All observations and model results are for 2001-2008 except at Neumayer (2001-2007). Note that the seasonal cycles are centered over local winter. The observed mean concentrations are indicated with filled black circles, while the lines are for the GEOS-Chem simulations (STD: black line, STD-SNOW: red line, STD-FF: green line). The black vertical lines and shaded areas correspond to the standard deviations of monthly means for observations and model simulations. For each individual panel, the legend lists mean concentrations and standard deviations, as well as the normalized mean bias (NMB=($\overline{\text{Model}}/\overline{\text{Obs}} - 1$)×100).**

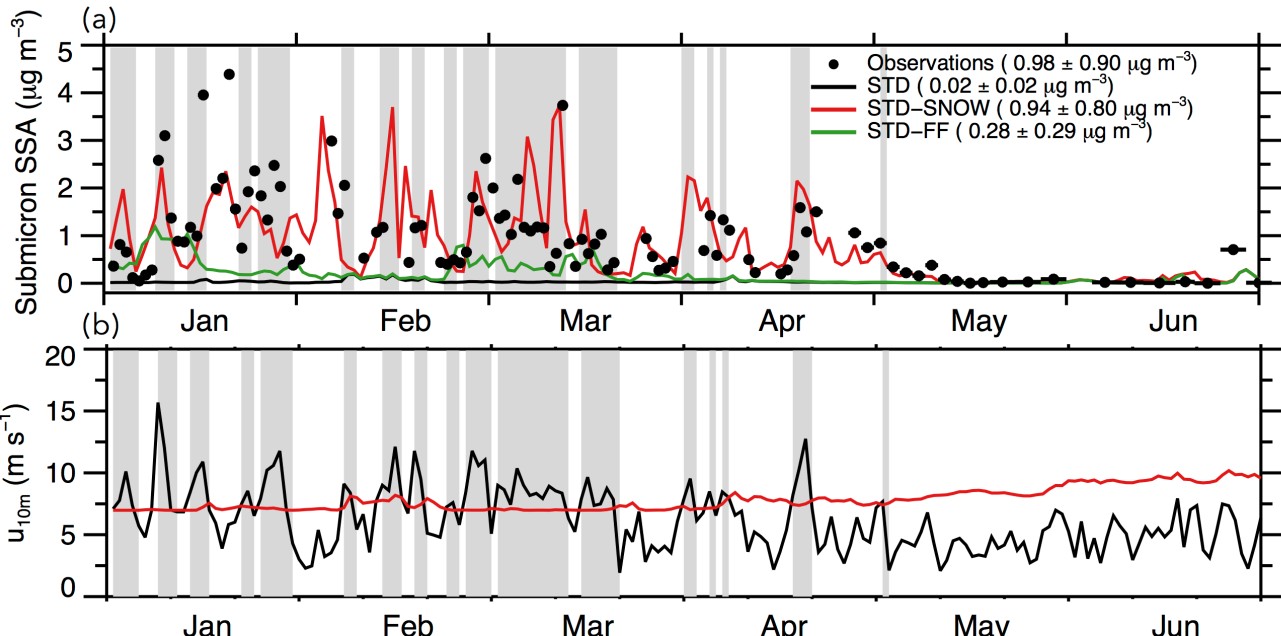

Figure 2: (a) Daily variations in submicron SSA mass concentrations at Barrow for 1 January to 30 June 2001. Observations are shown with filled black circles, while the GEOS-Chem simulations are indicated with lines (STD: black, STD-SNOW: red, and STD-FF: green). (b) MERRA 10m wind speed ($u_{10m}$) at Barrow. The red line indicates the wind speed threshold for blowing snow events calculated with the local MERRA 2m temperatures. Shaded gray areas indicate time periods when $u_{10m}$ exceeds the blowing snow wind threshold.

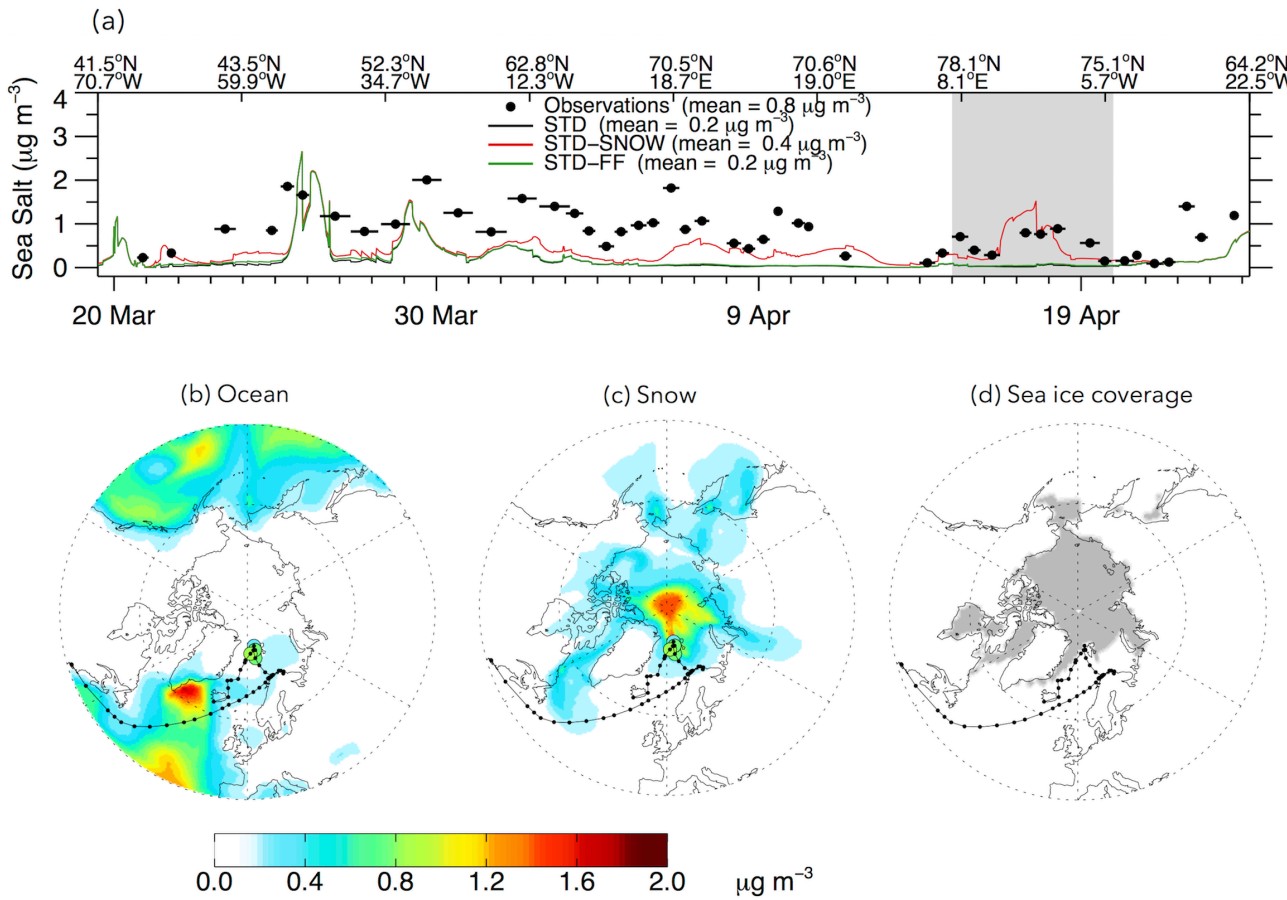

**Figure 3: (a) Timeseries of submicron SSA mass concentrations during the ICEALOT cruise between 19 March and 24 April 2008. Observations are shown as filled black circles with horizontal bars indicating the sampling period. The GEOS-Chem simulations are indicated with lines (STD: black, STD-SNOW: red, and STD-FF: green). The 15-19 April period discussed in the text is indicated by the gray shading. The bottom panels show the spatial distribution of mean surface SSA mass concentrations for the 15-19 April period. SSA mass concentrations due to open ocean emissions are shown in panel (b) while those due to blowing snow are shown in panel (c). The ship track is indicated with the black line and dots in panels b-d. The larger circles near Svalbard correspond to the location of the ship on 15-19 April, and they are color-coded based on observed SSA mass concentrations (same color scale are the model). Panel (d) displays the MERRA sea ice extent.**

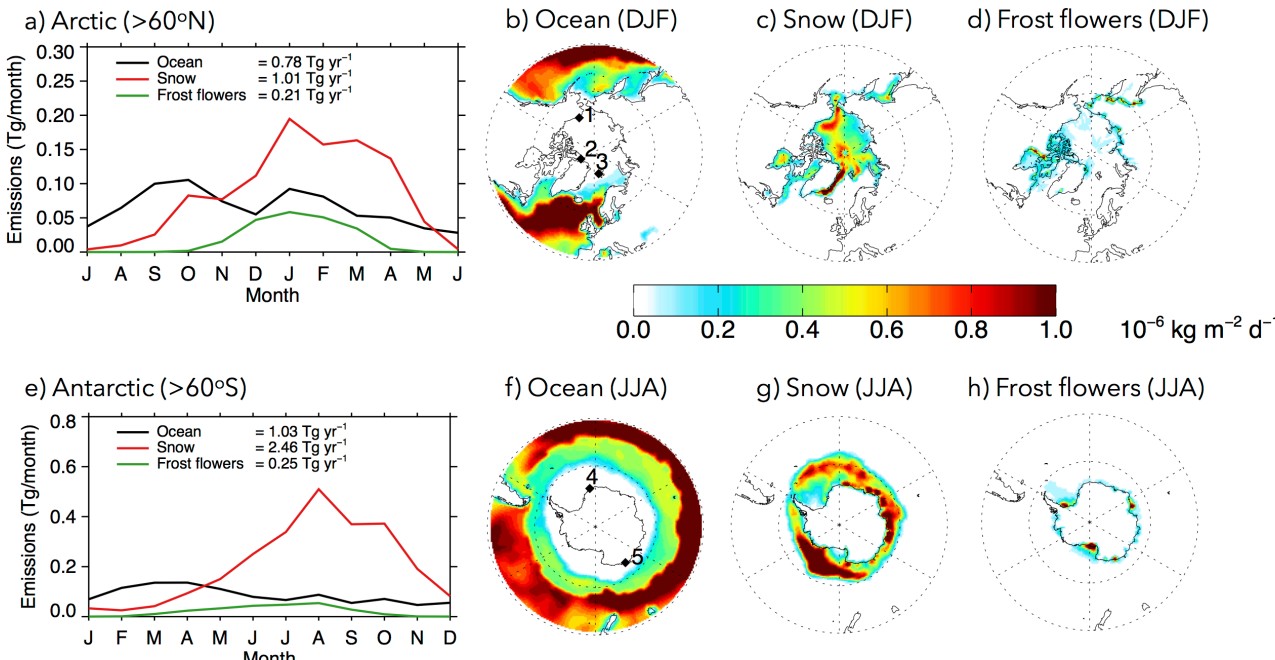

**Figure 4:** Seasonality of submicron SSA emissions in 2005 from open ocean, blowing snow, and frost flowers over (a) the Arctic and (e) the Antarctic for latitudes poleward of 60°. Also shown are spatial distributions of wintertime submicron SSA emissions over the Arctic (b-d) and the Antarctic (f-h). Filled diamonds in panels (b) and (f) correspond to the locations of Barrow (1), Alert (2), Zeppelin (3), Neumayer (4) and Dumont d'Urville (5).

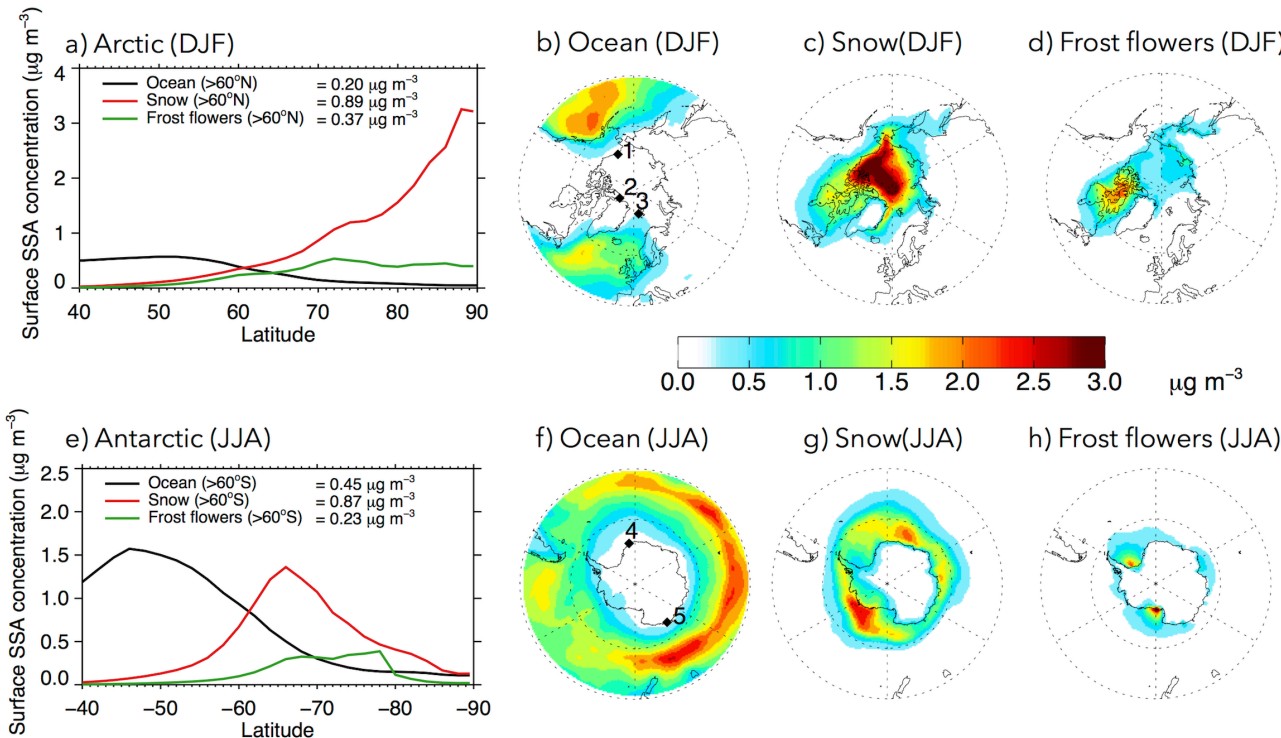

**Figure 5: Surface mass concentrations of wintertime submicron SSA. Zonal mean concentrations are shown over the (a) Arctic and (e) Antarctic for the open ocean (black line), blowing snow (red line), and frost flowers (green line). The panels on the right side show the spatial distributions of wintertime surface submicron SSA mass concentrations over the Arctic (b-d) and the Antarctic (f-h) for each source type. Filled diamonds in panels (b) and (f) correspond to the locations of the ground stations (see Fig. 3).**

