# Peer review of "Wintertime enhancements of sea salt aerosol in polar regions consistent with a sea-ice source from blowing snow"

_Atmospheric Chemistry and Physics, 2016_

## Referee Comment (RC1) · Anonymous Referee #1 · 16 Dec 2016

General Comments:

In this study, parameterizations for sea salt aerosol (SSA) emissions from blowing snow and frost flowers are implemented in the GEOS-Chem global chemical transport model. Model to measurement comparisons for SSA mass concentrations are presented for three Arctic and two Antarctic sites, as well as for an Arctic cruise. The authors conclude that blowing snow is a dominant SSA source during winter in the polar regions, with a smaller contribution from frost flowers. The paper is well written and addresses scientifically important questions regarding the sources for SSA in the polar regions. The related parameterizations are challenging to develop because there are several uncertainties involved. The impact of these uncertainties on the conclusions could be

discussed and examined more explicitly as outlined in the following comments. The manuscript should be suitable for publication if the following concerns can be satisfactorily addressed

Specific Comments:

1) P1, L12: The model with open ocean emissions alone underestimates the SSA mass concentrations by factors of 2-10. What is the uncertainty in the measurement SSA mass concentrations? Is it possible that the model and measurements could agree within the measurement uncertainty?

2) P2, L17-19: Please consider adding a sentence here to describe why the first two mechanisms lead to depletion of sulfate relative to sodium through the precipitation of mirabilite.

3) P4, L9: These lines note that the simulation with open water emissions alone of SSA overestimates the summertime SSA. Is it possible that the model could have errors in the summertime removal of SSA in precipitation in the polar regions (particularly if the model neglects aerosol removal by summertime drizzle). If the summertime removal is too inefficient, are you able to justify this suppression of the emissions? Are there any previous studies that have examined emissions for temperatures below 5 C? Fig. S1 was instructive since this seems to indicate a low sensitivity to this assumption about the emissions for the polar winter. Is there is a concern that this assumption might introduce a low bias to the open ocean emissions in an effort to compensate for errors in the removal scheme?

4) P4, L14-15: Are there any uncertainties related to the dry deposition parameterization? How might this affect your analysis, particularly in winter? As well, are there any uncertainties related to the wintertime removal by precipitation from mixed-phase and ice clouds and how might this affect your analysis?

5) P4, L18-19: Please consider clarifying here how the sodium bins are related to the

SSA bins in your parameterization. Are these sodium bins additional tracers in the model? As well, please specify where you mean by 'see below'.

6) P5, L26: In the frost flower parameterization, SSA emissions are only allowed if the wind speed is < 5 m s-1. However p. 2, lines 31-32 suggest that higher wind speeds are needed in order to break the frost flowers. There appears to be two processes here – the frost flowers form under low wind speeds, but do the emissions actually occur at greater wind speeds? Could this assumption that the frost flowers only emit SSA at low wind speed cause a low bias in the emissions from frost flowers? How do the emissions change if the frost flowers are allow to emit SSA at larger wind speeds than 5 m s-1, and how does this affect your conclusions?

7) P6, L12: A factor of 3.256 is used to convert all observed Na+ mass concentrations to SSA for comparison with the simulations. Is there any uncertainty in this factor that might make an apples-to-apples comparison between the measurements and model difficult?

8) P6, L12: 'use two size bins in the model' – are these bins for sea salt aerosol or sodium? Why are these bin limits different than in Table 1 and also different than P6, L21? As well, how are the observed Na+ concentrations in the submicron and supermicron ranges apportioned between the two bins used by the model for the purpose of the model-measurement comparison? What are the size limits for the observed submicron and supermicron aerosol?

9) P6, L23: Figure 1 shows a maximum in the measured SSA mass concentrations in the Arctic in November-December, whereas the simulations have a maximum Jan-Feb. What factors contribute to this model-measurement discrepancy?

10) Figure 1: The blowing snow simulation alone does appear to improve agreement with observations. However, in reality, both blowing snow and frost flowers might be expected to contribute together to the SSA concentrations. Have you conducted simulations with both of these sources implemented at the same time? Figure 1 seems to

suggest that if the model included both sources simultaneously, then the SSA would agree more closely with the observations in November/December in the Arctic and in April/May at Neumayer, but the model would strongly over predicts the observations in subsequent winter months. Please consider adding some related discussion. Does this suggest that the simulation might over predict the blowing snow emissions? As well, Fig. 3 seems to indicate that the model over predicts the SSA during the blowing snow event near 17 April.

11) P6, L24: As noted here, the advance and retreat of sea ice affects SSA. How well does the model simulation of sea ice agree with observations?

12) P7, L8-11: Why was the normalized mean bias chosen as the metric for the model evaluation as opposed to mean fractional bias? The latter metric has the advantages of not allowing a few data points to dominate the metric and allows for some error in measurements (Boylan and Russell 2006).

13) P9, L1-2: Blowing snow and frost flowers are noted to have a larger impact on the 0.01-0.5 um SSA mass concentrations than the open ocean source. What is the physical mechanism for relatively greater emission into this size range? For the case of blowing snow emissions, are there any uncertainties related to how the emitted mass is distributed between the two SSA bins using the assumed size distribution? Has the model been used for sensitivity studies to examine the potential impact of related uncertainties?

14) P9, L24: 'they are not efficient ice nuclei' – if possible, please add a reference to observations that support this statement. This seems in contradiction to some studies (e.g. DeMott et al. 2016). Or do you mean to indicate that SSA is not an efficient ice nucleus in the model?

15) P10, L20: How do these frost flower simulations compare to the recent work of Xu et al. (2016)?

[Figure]

16) P10, L28-30: As noted here, there are substantial uncertainties associated with these parameterizations. Since this is a model-based study, please consider whether the presentation of a few sensitivity study results related to the key uncertainties in the emissions parameterizations (salinity, size distribution for blowing snow and wind conditions for frost flower emissions) might be of help to the reader in interpreting the statement in the abstract that 'blowing snow is likely to be the dominant SSA source during the winter'. As well, this could help in interpreting the presented values for the SSA emissions from blowing snow and frost flowers since there seems to be some evidence that the blowing snow parameterization over predicts the observations.

17) Are you able to provide any recommendations to modelers about the implementation of blowing snow and frost flower parameterizations into global models?

18) P11, L9-10: For the potential impacts of wintertime SSA, would you expect any effect on mixed-phase and ice clouds?

References:

Boylan, J.W. and Russell, A. G.: PM and light extinction model performance metrics, goals, and criteria for three-dimensional air quality models. Atmos. Environ., 2006; 40, 4946-4959, doi:10.1016/j.atmosenv.2005.09.087.

DeMott P.J., Hill T.C.J., McCluskey C.S., et al.: Sea spray aerosol as a unique source of ice nucleating particles. Proceedings of the National Academy of Sciences of the United States of America. 2016; 113(21), 5797-5803, doi:10.1073/pnas.1514034112.

Xu, L., Russell, L.M. and Burrows, S.M.: Potential sea salt aerosol sources from frost flowers in the pan-Arctic region. J. Geophys. Res. Atmos., 2016; 121, 10,840–10,856, doi:10.1002/2015JD024713.

---

## Referee Comment (RC2) · Anonymous Referee #2 · 21 Dec 2016

This is a well-written paper that explores in a model the relative influence of three sources of sea salt aerosol in the polar regions. Although the blowing snow source has been explored in a model with a similar parameterisation in another model, this is the first time that all 3 sources have been tested in a similar setup. The paper compares model output with aerosol data at a number of polar sites, with rather impressive results in terms of concentration, seasonality and episodicity. It is able to conclude on the importance of the blowing snow source in wintertime, and provides reasonable evidence to dismiss the frost flower source as a significant player in most circumstances. Overall, it is a good paper, clear and well-argued, and certainly worth publishing in ACP. It provides a basis for exploring other aspects of the influence of different sources of sea

salt aerosol. My only substantial quibble with the authors is that set store by the fact that they get the concentrations right and that they are testing the balance between the sources. However this ignores the fact that they have had to take several decisions (such as the number of salt particles per snowflake, the salinity of the snow, and the scaling factor (Page 5, line 29), which are essentially tunings (ie they chose them in order to match the data). I think the paper should be a little clearer in recognising this, and in admitting that the relative strength of the different sources is influenced by this rather strongly. Detailed comments

Page 1, line 18. Here and elsewhere in the paper the authors refer to submicron aerosol, meaning the range which elsewhere they describe as the 0.01-0.50 $\mu$m radius range. It would be helpful if they would clearly state this usage, perhaps on page 4, line 17, where after describing the accumulation mode they could add "which we refer to as sub-micron based on its diameter".

Page 4, mid. I don't quite understand the description that for the blowing snow they treat just two size ranges. My understanding was that the Yang parameterisation that they are following uses many more size bins than that, and relies on this for many of its characteristics. Could the authors explain what they mean? Presumably they maintain different sizes in calculating the mass flux with respect to the number of snowflakes, so in what respect do they not use different size ranges and what are the likely impacts?

Page 5, line 5-6. I don't understand at all why they choose a lower salinity in the Antarctic, or why the higher Antarctic precipitation is relevant to that. It reads as if they think the snowfall somehow dilutes the salinity but this of course makes no sense as the salinity is more likely a function of snowpack thickness, which is likely lower in Antarctica. Please explain –at the moment this just looks likes a correction factor chosen at random.

Page 5, line 28. I think you mean "geometric mean diameter" and "geometric standard deviation". Geometric diameter doesn't seem meaningful.

Page 7, line 8. Please explain what M and O are; as written your explanation of what NMB is is unclear. I assume it's the percentage mismatch between model and data – why not call it model-data mismatch? The word "bias" seems wrong when you are simply comparing alternative partial sources to the data.

Page 9, line 22-25. I assume that you are saying that, for the same size and place the lifetime is the same whatever the source (this must be true), but that the lifetime is longer because the blowing snow sourced aerosol tends to form when it's colder. It might be clearer if you explain it more in this way.

Page 9, line 24. Just a question from my ignorance: wouldn't we expect sea salt to become an efficient ice nucleus below its eutectic (ie about 250K), when it would become a solid?

Fig 1 caption. Please explain here as well what NMB is: the reader should not have to read a quite difficult bit of text to understand the figure.

Fig 3 caption. I struggled to understand the text about the coloured circles. Do you mean that the larger circles (which are anyway hard to see) represent the ship's position between 15 and 19 April. If so, why not say this. If not, it needs a new explanation.

Fig 4 and 5 caption, just for clarity please add "submicron" in the phrase "spatial distributions of wintertime submicron SSA".

Supplement, section 1. Much of this text duplicates what is already written on page 4, para 1.

---

## Referee Comment (RC3) · Anonymous Referee #3 · 26 Dec 2016

This manuscript describes model-measurement comparison between GEOS-Chem with various sea-salt aerosol (SSA) sources and measurements from circum-Arctic field sites and a research ship cruise. The manuscript compares three models of SSA sources, open ocean, blowing snow, and frost flowers. Through the improved agreement between the model using the ocean + blowing snow model, the authors conclude that blowing snow is the dominant wintertime SSA source. The model using frost flowers is unable to match the observed seasonal cycles, and thus is indicated to be incorrect, and the model using only open ocean sources underpredicts SSA. The writing and logic of the manuscript are good, and the metrics for comparison are well defined and appropriate. Therefore, I support publication in ACP, with minor revisions.

General Comments:

1) The modeling in this manuscript uses a simple model for blowing snow based upon purely windspeed. However, experimental evidence (e.g. Sturm and Stuefer, 2013) shows that winds speed (alone) is insufficient to explain blowing snow fluxes fully.

Sturm, M. and Stuefer, S.: Wind-blown flux rates derived from drifts at arctic snow fences, J. Glaciol., 59, 21–34, doi: 10.3189/2013JoG12J110, 2013.

This experimental observation could be a partial explanation of deviations between the model and high time resolution data shown in Figure 2. However that citation also is not able to give a simple single equation for blowing snow, so is not a solution to this challenge, and the approach adopted by the authors is reasonable given the complexity.

2) The manuscript uses mass concentration data of SSA as the metric for model-measurement comparison. However, it would be useful to describe the rough size distribution of the modeled SSA and potentially some comparison between the model and observations. Table 1 shows this information, but it is only briefly discussed and it would be valuable to enhance the discussion. In addition, because SSA could be a source of cloud condensation nuclei, conversions of these numbers to number densities would also be valuable.

Specific Comments:

Page 1, line 18: The manuscript later defines "Over the Arctic..." as >60 degrees. these statements (in the abstract) should also include the definition.

Page 2, line 2: I would say "waters are mostly covered by sea ice."

Page 2, line 11: It would be more accurate here to describe studies that support SSA formation from frost flowers, as well as ones that don't support frost-flower SSA. The section later (at the bottom of this page and top of next) contains the references that are relevant.

[Figure]

Page 4, line 28: I believe that other measurements of Arctic surface snow could be compared to the 0.1 PSU concentration. Toom-Sauntry and Barrie (2002) measured fresh snowfall, and Krnavek et al. (2012) have fairly extensive data sets.

Toom-Sauntry, D. and Barrie, L. A.: Chemical composition of snowfall in the high Arctic: 1990 – 1994, Atmos. Environ., 36, 2683–2693, doi:10.1016/S1352-2310(02)00115-2, 2002.

Krnavek, L., Simpson, W. R., Carlson, D., Domine, F., Douglas, T. A., and Sturm, M.: The chemical composition of surface snow in the Arctic: Examining marine, terrestrial, and atmospheric influences, Atmos. Environ., 50, 349–359, 2012.

Page 5, line 9: "ease" instead of "easiness"

Page 7, around line 12: It would be useful to mention that modeled spatial maps will be presented later.

Page 8, line 30: The wording here is a bit confusing, because the normal conditions used for modeling frost flower formation include open water and cold temperatures. In this work, page 5, line 26 indicates that frost flowers are suppressed by the high winds (or are covered by drifting snow), which is the origin of the statement. Please reword this section to indicate clearly that the high winds (rather than open water and cold temperatures) are the reason for "inhibition of frost flowers".

---

## Author Comment (AC1) · 21 Feb 2017

We would like to thank referee #1 for very helpful comments and insightful suggestions.

**Reply to comments by Anonymous Referee #1**

General Comments:
In this study, parameterizations for sea salt aerosol (SSA) emissions from blowing snow and frost flowers are implemented in the GEOS-Chem global chemical transport model. Model to measurement comparisons for SSA mass concentrations are presented for three Arctic and two Antarctic sites, as well as for an Arctic cruise. The authors conclude that blowing snow is a dominant SSA source during winter in the polar regions, with a smaller contribution from frost flowers. The paper is well written and addresses scientifically important questions regarding the sources for SSA in the polar regions. The related parameterizations are challenging to develop because there are several uncertainties involved. The impact of these uncertainties on the conclusions could be discussed and examined more explicitly as outlined in the following comments. The manuscript should be suitable for publication if the following concerns can be satisfactorily addressed.

Specific Comments
1) P1, L12: The model with open ocean emissions alone underestimates the SSA mass concentrations by factors of 2-10. What is the uncertainty in the measurement SSA mass concentrations? Is it possible that the model and measurements could agree within the measurement uncertainty?

- The reported uncertainties in the measurements of $Na^+$ mass concentrations are 11% (relative uncertainty, 0.01 $\mu g/m^3$) at Barrow (Quinn et al., 2000), 5-10% (0.01 $\mu g/m^3$) at Alert and Zeppelin (WMO/GAW, 2003), 5-11% at Neumayer (Weller et al. 2008), and 5% at Dumont d'Urville (Legrand et al. 2012). This should have been included in the original manuscript, thank you for pointing out this omission. As observed wintertime $Na^+$ concentrations are ~0.3-1 $\mu g/m^3$, factors of 2-10 disagreement are well outside the observational uncertainties. We have added a discussion of measurement uncertainties in section 2.3:
"The $Na^+$ mass concentrations are determined by ion chromatography with uncertainties of 5%-11% (0.01 $\mu g/m^3$ in absolute uncertainty)."

2) P2, L17-19: Please consider adding a sentence here to describe why the first two mechanisms lead to depletion of sulfate relative to sodium through the precipitation of mirabilite.

-We have clarified this in the revised manuscript:
"The first two of these mechanisms lead to depletion of the sulfate to sodium ratio relative to bulk sea water as mirabilite ($Na_2SO_4 \cdot 10H_2O$) precipitates from brine at temperatures below -8°C during sea ice and frost flower formation (Alvarez-Aviles et al., 2008)."

3) P4, L9: These lines note that the simulation with open water emissions alone of SSA overestimates the summertime SSA. Is it possible that the model could have errors in the summertime removal of SSA in precipitation in the polar regions (particularly if the model neglects aerosol removal by summertime drizzle). If the summertime removal is too inefficient, are you able to justify this suppression of the emissions? Are there any previous studies that have examined emissions for temperatures below 5 C? Fig. S1 was instructive since this seems to indicate a low sensitivity to this assumption about the emissions for the polar winter. Is there is a concern that this assumption might introduce a low bias to the open ocean emissions in an effort to compensate for errors in the removal scheme?

- We agree with the referee that an alternative explanation for the model overestimate of summertime SSA could be related to an underestimate in the removal of SSA during summer. This point has been noted in the revised manuscript. There is support for our cold temperature suppression assumption from the laboratory study by Mårtensson et al. (2003). They examined marine aerosol production from seawater at -2$^o$C and 5$^o$C, finding a 50-60% decrease in the number of particles produced for r>0.1µm when temperatures decrease from 5$^o$C to -2$^o$C. This is consistent with our assumption of suppression at very cold temperatures.
As noted by the referee, this modification to open ocean emissions does not affect our wintertime results, as the five polar sites are distant from open ocean, with very small amounts of open ocean SSA transported to those sites. For example, at Barrow and Alert wintertime SSA concentrations due to the open ocean are <0.1 µg/m$^3$, more than an order of magnitude lower than observations (1-3 µg/m$^3$).

We have added the following:
"This is consistent with the laboratory study of Mårtensson et al. (2003), who report a 50-60% decrease in aerosol production (for r>0.1 µm) when seawater temperature decreased from 5$^o$C to -2$^o$C. We note that another potential explanation for the summertime overestimate in SSA mass concentrations is inefficient wet removal from low-intensity summer precipitation in GEOS-Chem (Croft et al., 2016)."

4) P4, L14-15: Are there any uncertainties related to the dry deposition parameterization? How might this affect your analysis, particularly in winter? As well, are there any uncertainties related to the wintertime removal by precipitation from mixed-phase and ice clouds and how might this affect your analysis?

- The dry deposition velocity for aerosols over snow and ice is assumed to be 0.03 cm/s in the model, which is based on the estimates from measurements in the Arctic. As discussed in the manuscript (p10, L2), for cold clouds (mixed-phase and ice clouds with temperature < 258 K) SSA is only removed by below-cloud precipitation and we assume no in-cloud scavenging in the model. If in-cloud scavenging were to be an efficient process for SSA, it would lead to decreases in our calculated SSA mass concentrations and require a larger source from sea ice to match observed concentrations. Previous studies using the GEOS-Chem model found reasonable agreement with ground-based and aircraft observations of sulfate, ammonium, black carbon, and organic aerosols

during winter and spring (Fisher et al., 2011; Wang et al., 2011). This suggests that the removal processes during winter and spring are reasonably represented in the model. This is now noted in the revised manuscript.

"Detailed comparisons of GEOS-Chem black carbon and organic aerosol (Wang et al., 2011) as well as sulfate and ammonium aerosol (Fisher et al., 2011) to ground-based and aircraft observations over the Arctic during winter and spring suggest that transport and removal processes are reasonable captured by the model. "

5) P4, L18-19: Please consider clarifying here how the sodium bins are related to the SSA bins in your parameterization. Are these sodium bins additional tracers in the model? As well, please specify where you mean by 'see below'.

- We agree that this was unclear and have clarified this in the revised manuscript:

"For this work, we track SSA mass in two size bins: accumulation mode ($r_{dry} = 0.01-0.5$ μm) and coarse mode ($r_{dry} = 0.5-4$ μm), except in the comparison to in situ mass concentrations of SSA for which we use $r_{dry} = 0.01-0.3$ μm and $r_{dry} = 0.3-3$ μm (see section 2.3). "

6) P5, L26: In the frost flower parameterization, SSA emissions are only allowed if the wind speed is < 5 m s$^{-1}$. However p. 2, lines 31-32 suggest that higher wind speeds are needed in order to break the frost flowers. There appears to be two processes here – the frost flowers form under low wind speeds, but do the emissions actually occur at greater wind speeds? Could this assumption that the frost flowers only emit SSA at low wind speed cause a low bias in the emissions from frost flowers? How do the emissions change if the frost flowers are allow to emit SSA at larger wind speeds than 5 m s$^{-1}$, and how does this affect your conclusions?

- We recognize that there is likely to be a distinction between frost flower formation under low windspeed and lofting at high windspeed. Given the uncertainty and contradictory results in the literature regarding frost flower formation and lofting (see our section 1), we have conducted a frost flower simulation removing the wind speed limit and reverting to the formulation proposed by Xu et al. (2013). This also enables us to address this referee's comment #15 about the more recent study of Xu et al. (2016). We compare in the figure below our original frost flower simulation (STD-FF green line) to the one with no wind speed limit (STD-FF-noWindLimit purple line). As we use the same scaling factor as Xu et al. (2013), the new emissions from frost flowers are slightly lower than in our STD-FF simulation. We find good agreement at Alert for Dec-Mar, and at Barrow in January-February, but for all the other stations our results remain unchanged, with very small influence of frost flowers at Zeppelin, Neumayer and Dumont d'Urville, and therefore we find that frost flower emissions are insufficient to explain the wintertime SSA enhancements in the observations at these sites. Thus overall our conclusions remain unchanged.

For simplicity, we replace our original FF simulation with this new updated one, without any windspeed limit. The figures and text have been changed to reflect this.

[Figure]

7) P6, L12: A factor of 3.256 is used to convert all observed Na$^+$ mass concentrations to SSA for comparison with the simulations. Is there any uncertainty in this factor that might make an apples-to-apples comparison between the measurements and model difficult?

- This mass ratio for Na$^+$ to SSA (3.256) is based on the standard seawater mass ratio at a salinity of 35 psu. There can be some small changes to the ionic composition of SSA from frost flowers and snow, but these are small (<0.5%). Based on the chemical composition of frost flowers reported in Rankin et al. (2000), this ratio is 3.237 for frost flowers. Krnavek et al. (2012) find that this mass ratio is 3.278 for snow on thick first year sea ice and 3.24 for those on thin ones. Therefore, we think it is reasonable to use a constant factor (=3.256) to convert observed Na$^+$ to SSA for comparison with the model SSA.

We have added this to the revised manuscript:
"For comparison between the GEOS-Chem model and the observations, we convert observed Na$^+$ mass concentrations to SSA mass concentrations using a factor of 3.256

based on the mass ratio of Na$^+$ in seawater (Riley and Chester, 1971). For frost flowers, this ratio is 3.237 (Rankin et al., 2000). Krnavek et al. (2012) find a ratio of 3.24-3.278 for snow on first year sea ice. As this ratio varies by less than 0.5% for these different SSA sources, we use a constant factor of 3.256."

8) P6, L12: 'use two size bins in the model' – are these bins for sea salt aerosol or sodium? Why are these bin limits different than in Table 1 and also different than P6, L21? As well, how are the observed Na$^+$ concentrations in the submicron and super-micron ranges apportioned between the two bins used by the model for the purpose of the model-measurement comparison? What are the size limits for the observed submicron and supermicron aerosol?

- Yes, this indeed requires some clarification. These bins are dry sea salt aerosol radii. For simplicity, we now use one single set of size bins (rdry=0.01-0.3 μm and 0.3-3 μm) for comparison to the ground sites and ICEALOT cruises and justify this in the revised text. Most previously published studies use (0.01-0.5 μm; 0.5-4 μm), so this is why we use these bins in Table 1.

"The reported aerodynamic cut-off diameters of the measurements are 1 μm and 10 μm at Barrow and during ICEALOT (Quinn et al., 2002), and 7-10 μm at the other sites (Wagenbach et al., 1998; WMO/GAW, 2003; Weller et al., 2008). In order to compare to model simulations, we need to convert these aerodynamic diameters to dry geometric radii. This conversion depends on aerosol density, relative humidity during sampling, and whether the particle is spherical (Seinfeld and Pandis, 2006). For example a 10 μm aerodynamic diameter could correspond to a dry geometric radius of 2.3 μm (80% RH, 1.2 g/cm$^3$ pure NaCl solution, factor of 2 growth between dry and 80% RH), 3 μm (dry cubical NaCl particle, 2.2 g/cm$^3$, Lewis and Schwartz, 2004), 3.8 μm (30% RH, ammonium sulfate and sea salt aerosol, 1.7 g/cm$^3$, Quinn et al., 1996). Thus for comparison to observations we conduct a simulation with two size bins: $r_{dry} = 0.01-0.3$ μm and $r_{dry} = 0.3-3$ μm."

9) P6, L23: Figure 1 shows a maximum in the measured SSA mass concentrations in the Arctic in November-December, whereas the simulations have a maximum Jan-Feb. What factors contribute to this model-measurement discrepancy?

- Given the simplified nature of the blowing snow simulation, a number of potential factors could explain this offset, such as seasonal and spatial variability in snow mobility and/or salinity. Exploring these factors is beyond the scope of our manuscript.

10) Figure 1: The blowing snow simulation alone does appear to improve agreement with observations. However, in reality, both blowing snow and frost flowers might be expected to contribute together to the SSA concentrations. Have you conducted simulations with both of these sources implemented at the same time? Figure 1 seems to suggest that if the model included both sources simultaneously, then the SSA would agree more closely with the observations in November/December in the Arctic and in April/May at Neumayer, but the model would strongly over predicts the observations in

subsequent winter months. Please consider adding some related discussion. Does this suggest that the simulation might over predict the blowing snow emissions? As well, Fig. 3 seems to indicate that the model over predicts the SSA during the blowing snow event near 17 April.

- It is indeed possible that both blowing snow and frost flowers contribute together to SSA concentrations. We have included a new figure in the supplemental material (Fig S5) where we compare a simulation with all 3 sources (open ocean, frost flowers and blowing snow) to observations. The resulting simulation overestimates observations, especially at Barrow and Alert, where modeled SSA is a factor of 2-3 too high in January-March. At the other 3 sites the influence of frost flowers is small, so the differences are not as large.

We have added this following discussion in P9, L1-3.

"It is possible that both blowing snow and frost flower emissions act together. However, when we add the contributions from both sources, we find that modelled SSA mass concentrations are a factor of 2-3 too high compared to observations at Barrow and Alert (Fig. S5). In particular, the frost flower simulation leads to a peak in SSA in February at Barrow, which is not observed."

11) P6, L24: As noted here, the advance and retreat of sea ice affects SSA. How well does the model simulation of sea ice agree with observations?

- As described in Section 2.1, the sea ice in MERRA is set as a boundary condition from the NOAA Optimal Interpolation (OI) v2 gridded dataset (Reynolds et al., 2002). This dataset is based on SSMI satellite observations, so should have a reasonable representation of sea ice cover. This has been clarified in section 2.1:

"The sea ice concentration boundary conditions in MERRA are derived from the weekly product of Reynolds et al. (2002), which is based on Special Sensor Microwave Imager (SSMI) instruments on Defense Meteorological Satellite Program (DMSP) satellites."

12) P7, L8-11: Why was the normalized mean bias chosen as the metric for the model evaluation as opposed to mean fractional bias? The latter metric has the advantages of not allowing a few data points to dominate the metric and allows for some error in measurements (Boylan and Russell 2006).

-We use the normalized mean bias (NMB=$(\overline{Model}/\overline{Obs} - 1)\times100$) here as we assume that the observations are the absolute truth for evaluating the performance of different model simulations. We are using multi-year (7-8 years) observations, which helps decrease the influence of a few daily/weekly points.

13) P9, L1-2: Blowing snow and frost flowers are noted to have a larger impact on the 0.01-0.5 um SSA mass concentrations than the open ocean source. What is the physical

mechanism for relatively greater emission into this size range? For the case of blowing snow emissions, are there any uncertainties related to how the emitted mass is distributed between the two SSA bins using the assumed size distribution? Has the model been used for sensitivity studies to examine the potential impact of related uncertainties?

- The physical mechanisms for SSA emissions from open ocean emissions (breaking waves and bubble bursting) are different than emissions from blowing snow and frost flowers (saltation of fallen snow and suspension) or frost flowers (saltation of broken frost flower crystals and suspension). The original crystalline form of snow particles/frost flower fragments are expected to be shattered by repeated impacts with the ground and other particles during saltation. For blowing snow, the size of the dry SSA particle depends on the original snow particle size, snow salinity, and number of SSA particles produced per snow particle (Yang et al., 2008). The blowing snow particle size distribution is constrained by observations of blowing snow events (Mann et al., 2000). As discussed in the text, there are few observations of salinity of snow on sea-ice so we used uniform values of salinity for the entire Arctic and Antarctic snow salinity. This is certainly something that can be improved as more measurements become available. Assumptions of the number SSA particles produced per snow particle are highly uncertain, and we include a sensitivity study in the supplement (Fig. S2).

We have added the following in the revised manuscript:
"This difference in size distributions is related to the different physical mechanisms for SSA emissions from open ocean emissions (breaking waves and bubble bursting) compared to blowing snow (saltation of fallen snow and suspension) or frost flowers (saltation of broken frost flower crystals and suspension). The original crystalline form of snow particles/frost flower fragments are expected to be shattered by repeated impacts with the ground and other particles during saltation. Sublimation of ice from these particles leads to relatively small SSA compared to bubble bursting in the open ocean."

14) P9, L24: 'they are not efficient ice nuclei' – if possible, please add a reference to observations that support this statement. This seems in contradiction to some studies (e.g. DeMott et al. 2016). Or do you mean to indicate that SSA is not an efficient ice nucleus in the model?

- Excellent point. This statement is indeed based on what is assumed in the GEOS-Chem model for scavenging. We clarify this in the text and add references to DeMott et al. (2016) and Wise et al. (2012), and note that in-cloud scavenging is not currently included in GEOS-Chem.

"Recent laboratory studies have shown that SSA could act as ice nuclei by deposition freezing (Wise et al., 2012) and immersion freezing (DeMott et al., 2016), and might thus undergo in-cloud scavenging in mixed and ice clouds. This process is not currently included in GEOS-Chem."

15) P10, L20: How do these frost flower simulations compare to the recent work of Xu et al. (2016)?

- Thank you for bring this work to our attention. Both the magnitude of emissions and spatial distribution of our frost flower simulation are similar to the results Xu et al. (2016) for the Arctic. This has been added in the revised manuscript:

"Our emissions from frost flowers over the Arctic (0.21 Tg/yr) are consistent the accumulation model emissions reported by Xu et al. (2016) (0.24 Tg for November-February, their Table 2). We also find a similar geographic distribution."

16) P10, L28-30: As noted here, there are substantial uncertainties associated with these parameterizations. Since this is a model-based study, please consider whether the presentation of a few sensitivity study results related to the key uncertainties in the emissions parameterizations (salinity, size distribution for blowing snow and wind conditions for frost flower emissions) might be of help to the reader in interpreting the statement in the abstract that 'blowing snow is likely to be the dominant SSA source during the winter'. As well, this could help in interpreting the presented values for the SSA emissions from blowing snow and frost flowers since there seems to be some evidence that the blowing snow parameterization over predicts the observations.

- Yes, we agree with the referee that there are substantial uncertainties in the parameterizations and have clarified the text, as well as added sensitivity simulations (wind threshold for frost flowers) in addition to the sensitivity simulation for the blowing snow particle number. We now note in the revised text:

"One key uncertainty in our simulations is snow salinity. Indeed, SSA emissions from blowing snow have a near-linear dependence on the salinity of snow. Thus a doubling of the assumed salinity would lead to a doubling in SSA emissions from blowing snow. Furthermore, we assume a uniform salinity of snow over both first-year and multi-year sea ice. This likely overestimates the contribution of blowing snow SSA over the western Arctic, which is dominated by multi-year sea ice. More extensive observations of surface snow salinity at multiple locations over both first-year and multi-year sea ice can help further refine these assumptions. Sampling of SSA size distributions during blowing snow events can help determine the number of particles per snowflake, which we determined empirically in this study. This number will not affect to total SSA emissions, but will change the relative importance of submicron and supermicron SSA emissions."

We have added a sensitivity simulation with/without wind conditions of frost flower emission is detailed in the supplementary material.

The overestimate of SSA in spring by the blowing model may be due to the seasonal variability of the surface snow salinity which is not taken into account in this study. This should be further constrained with observations of surface snow salinity at different location and times which are limited at this moment.

17) Are you able to provide any recommendations to modelers about the implementation of blowing snow and frost flower parameterizations into global models?

- Our implementation of blowing snow follows Yang et al. (2008) with modification of salinity, age of snow and number of particles. And the implementation of frost flower emission follows the emission scheme in Xu et al. (2013). Details on implementation of these sources in the model can be found in section 2.2, Yang et al. (2008), Xu et al. (2013) and references therein.

18) P11, L9-10: For the potential impacts of wintertime SSA, would you expect any effect on mixed-phase and ice clouds?

-Yes, this is indeed a very good point. We have added the following statement:

[revised manuscript text omitted]

---

## Author Comment (AC3) · 21 Feb 2017

We thank referee #3 for their helpful review and address the comments below.

**Reply to comments by Anonymous Referee #3**

This manuscript describes model-measurement comparison between GEOS-Chem with various sea-salt aerosol (SSA) sources and measurements from circum-Arctic field sites and a research ship cruise. The manuscript compares three models of SSA sources, open ocean, blowing snow, and frost flowers. Through the improved agreement between the model using the ocean + blowing snow model, the authors conclude that blowing snow is the dominant wintertime SSA source. The model using frost flowers is unable to match the observed seasonal cycles, and thus is indicated to be incorrect, and the model using only open ocean sources underpredicts SSA. The writing and logic of the manuscript are good, and the metrics for comparison are well defined and appropriate. Therefore, I support publication in ACP, with minor revisions.

General Comments:

1) The modeling in this manuscript uses a simple model for blowing snow based upon purely windspeed. However, experimental evidence (e.g. Sturm and Stuefer, 2013) shows that winds speed (alone) is insufficient to explain blowing snow fluxes fully.

Sturm, M. and Stuefer, S.: Wind-blown flux rates derived from drifts at arctic snow fences, J. Glaciol., 59, 21–34, doi: 10.3189/2013JoG12J110, 2013.

This experimental observation could be a partial explanation of deviations between the model and high time resolution data shown in Figure 2. However that citation also is not able to give a simple single equation for blowing snow, so is not a solution to this challenge, and the approach adopted by the authors is reasonable given the complexity.

-We agree with the referee that other factors, in addition to windspeed, are likely to affect the blowing snow flux. We choose to use the Yang et al. (2008) parameterization, which itself is based on the blowing snow sublimation parameterization of Déry and Yau (2001), as this seems to be a well-established formulation.

2) The manuscript uses mass concentration data of SSA as the metric for model-measurement comparison. However, it would be useful to describe the rough size distribution of the modeled SSA and potentially some comparison between the model and observations. Table 1 shows this information, but it is only briefly discussed and it would be valuable to enhance the discussion. In addition, because SSA could be a source of cloud condensation nuclei, conversions of these numbers to number densities would also be valuable.

- We have added the following discussion in p5, L20:
"Our simulation yields blowing snow SSA emissions with 38% of SSA mass in the submicron range (0.01−0.5 μm) and 62% in the supermicron range (0.01−0.5 μm) for the Arctic. As we assume a lower salinity in the Antarctic, more of the blowing snow emissions are in the submicron range (60%) in that region."

As we do not track number concentrations in the model, we cannot comment on the resulting CCN concentrations. Based on this referee's comment and the other referees as well we have added more discussion on the potential role of SSA as ice nuclei.

Page 1, line 18: The manuscript later defines "Over the Arctic..." as >60 degrees. these statements (in the abstract) should also include the definition.

-We have made the change in the manuscript.

Page 2, line 2: I would say "waters are mostly covered by sea ice."

-We have made the change in the manuscript.

Page 2, line 11: It would be more accurate here to describe studies that support SSA formation from frost flowers, as well as ones that don't support frost-flower SSA. The section later (at the bottom of this page and top of next) contains the references that are relevant.

- Good point. We have moved the paragraph describing studies that don't support SSA formation from frost flowers higher up.

Page 4, line 28: I believe that other measurements of Arctic surface snow could be compared to the 0.1 PSU concentration. Toom-Sauntry and Barrie (2002) measured fresh snowfall, and Krnavek et al. (2012) have fairly extensive data sets.

Toom-Sauntry, D. and Barrie, L. A.: Chemical composition of snowfall in the high Arctic: 1990 – 1994, Atmos. Environ., 36, 2683–2693, doi:10.1016/S1352-2310(02)00115-2, 2002.

Krnavek, L., Simpson, W. R., Carlson, D., Domine, F., Douglas, T. A., and Sturm, M.: The chemical composition of surface snow in the Arctic: Examining marine, terrestrial, and atmospheric influences, Atmos. Environ., 50, 349–359, 2012.

-We thank this referee to pointing out these studies. We have added a discussion of these measurements in the revised manuscript.
 "This profile is consistent with a salinity source from the underlying sea ice and little influence from atmospheric deposition. Toom-Sauntry and Barrie (2002) find that freshly fallen snow itself tends to have low salinity (<0.01 psu). For simplicity, we assume the same salinity for surface snow on first-year and multi-year sea ice, although we recognize that in reality the surface snow may be less salty on multi-year sea ice due to less efficient upward transport of brine. Indeed, Krnavek et al. (2012) reported that the ion concentrations of surface snow sampled in the Alaskan Arctic display large variability depending on sea ice type: 0.01 psu for snow on multiyear sea ice, 0.1 psu for snow on thick first year sea ice and 0.8 psu for snow on thin first year sea ice. "

Page 5, line 9: "ease" instead of "easiness"

-We have made the change in the manuscript.

Page 7, around line 12: It would be useful to mention that modeled spatial maps will be presented later.

-We have made the change in the manuscript.

Page 8, line 30: The wording here is a bit confusing, because the normal conditions used for modeling frost flower formation include open water and cold temperatures. In this work, page 5, line 26 indicates that frost flowers are suppressed by the high winds (or are covered by drifting snow), which is the origin of the statement. Please reword this section to indicate clearly that the high winds (rather than open water and cold temperatures) are the reason for "inhibition of frost flowers".

-We have removed this statement from our manuscript, as based on referee #1's comments we now use a frost flower simulation with no wind inhibition. The wind inhibition simulation is included in the supplementary material.

**Reference**

Déry, S. J. and Yau, M. K.: Simulation of blowing snow in the Cana- dian Arctic using a double-moment model, Bound.-Lay. Meteo- rol., 99, 297–316, 2001.

Krnavek, L., Simpson, W. R., Carlson, D., Domine, F., Douglas, T. A., and Sturm, M.: The chemical composition of surface snow in the Arctic: Examining marine, terrestrial, and atmospheric influences, Atmos. Environ., 50, 349–359, 2012.

Toom-Sauntry, D. and Barrie, L. A.: Chemical composition of snowfall in the high Arctic: 1990 – 1994, Atmos. Environ., 36, 2683–2693, doi:10.1016/S1352-2310(02)00115-2, 2002.

Yang, X., Pyle, J. A., and Cox, R. A.: Sea salt aerosol production and bromine release: Role of snow on sea ice, Geophys. Res. Lett., 35, L16815, doi:10.1029/2008GL034536, 2008.

---

## Author Comment (AC2)

We would like to thank referee #2 for their review and excellent suggestions.

**Reply to comments by Anonymous Referee #2**

This is a well-written paper that explores in a model the relative influence of three sources of sea salt aerosol in the polar regions. Although the blowing snow source has been explored in a model with a similar parameterisation in another model, this is the first time that all 3 sources have been tested in a similar setup. The paper compares model output with aerosol data at a number of polar sites, with rather impressive results in terms of concentration, seasonality and episodicity. It is able to conclude on the importance of the blowing snow source in wintertime, and provides reasonable evidence to dismiss the frost flower source as a significant player in most circumstances. Overall, it is a good paper, clear and well-argued, and certainly worth publishing in ACP. It provides a basis for exploring other aspects of the influence of different sources of sea salt aerosol. My only substantial quibble with the authors is that set store by the fact that they get the concentrations right and that they are testing the balance between the sources. However this ignores the fact that they have had to take several decisions (such as the number of salt particles per snowflake, the salinity of the snow, and the scaling factor (Page 5, line 29), which are essentially tunings (ie they chose them in order to match the data). I think the paper should be a little clearer in recognising this, and in admitting that the relative strength of the different sources is influenced by this rather strongly.

 - We have added more discussion of our assumptions in the revised text and have recognized more clearly how these assumptions impact our conclusions. In particular, we have added the following in our conclusions:

"The SSA parameterizations for blowing snow and frost flowers have several intrinsic assumptions, such as the salinity of snow and the scaling factor for frost flowers, which influence the relative magnitudes of these two sources in polar regions. The geographic distribution, seasonal cycle, and daily variability of these sources, however, is controlled by sea ice extent and meteorological parameters (winds and temperature). In this study, we showed that the temporal and geographical variability of SSA observations at five polar sites is more consistent with blowing snow than with frost flowers. Based on this comparison, we conclude that blowing snow is likely to be the dominant source of SSA in polar winter, although frost flowers cannot be entirely ruled out. In particular, they may contribute indirectly to SSA emissions by salinating wind-blown snow (Obbard et al., 2009)."

Detailed comments

Page 1, line 18. Here and elsewhere in the paper the authors refer to submicron aerosol, meaning the range which elsewhere they describe as the 0.01-0.50 μm radius range. It would be helpful if they would clearly state this usage, perhaps on page 4, line 17, where after describing the accumulation mode they could add "which we refer to as sub-micron based on its diameter".

- This has been clarified in the manuscript:

"For this work, we track SSA mass in two size bins: accumulation mode ($r_{dry}$ = 0.01−0.5 µm) and coarse mode ($r_{dry}$ = 0.5–4 µm), except in the comparison to in situ mass concentrations of SSA for which we use $r_{dry}$ = 0.01−0.3 µm and $r_{dry}$ = 0.3−3 µm (see section 2.3). In the rest of the manuscript we will refer to the accumulation and coarse mode SSA aerosol as submicron and supermicron SSA based on their diameters."

Page 4, mid. I don't quite understand the description that for the blowing snow they treat just two size ranges. My understanding was that the Yang parameterisation that they are following uses many more size bins than that, and relies on this for many of its characteristics. Could the authors explain what they mean? Presumably they maintain different sizes in calculating the mass flux with respect to the number of snowflakes, so in what respect do they not use different size ranges and what are the likely impacts?

- We use the same size distribution for snow particles as reported in Yang et al. (2008), but use a different salinity and number of SSA particle per snowflake. As we track SSA mass in GEOS-Chem in two size bins, we integrate this size distribution and calculate the corresponding SSA emissions for the two size bins.

Page 5, line 5-6. I don't understand at all why they choose a lower salinity in the Antarctic, or why the higher Antarctic precipitation is relevant to that. It reads as if they think the snowfall somehow dilutes the salinity but this of course makes no sense as the salinity is more likely a function of snowpack thickness, which is likely lower in Antarctica. Please explain –at the moment this just looks likes a correction factor chosen at random.

-This point was not clearly described in the original manuscript and we have clarified this in the revised version. Yes indeed, it is the snow thickness that controls the salinity of the snow on sea ice. Antarctic is surrounded by ocean, so precipitation occurs more frequently over Antarctic sea ice. As a result, snowpack tends to be thick. The Arctic Ocean is surrounded by land and precipitation is relatively rare, hence the snowpack is thinner on Arctic sea ice.

"Snowpack on Antarctic sea ice is thicker than over Arctic sea ice. Indeed, Antarctica is surrounded by the Southern Ocean which brings moisture, while the Arctic is surrounded by land with wintertime precipitation that is 3 times lower than over Antarctic sea ice (Huffman et al., 2001). As the salinity of snow decreases with snowpack thickness, we assume that the salinity of snow on Antarctic sea ice is 0.03 psu, a factor of 3 lower than over the Arctic following Yang et al. (2010). "

Page 5, line 28. I think you mean "geometric mean diameter" and "geometric standard deviation". Geometric diameter doesn't seem meaningful.

- Yes, this was a mistake. We have corrected this in the revised manuscript.

Page 7, line 8. Please explain what M and O are; as written your explanation of what NMB is is unclear. I assume it's the percentage mismatch between model and data − why not call it model-data mismatch? The word "bias" seems wrong when you are simply comparing alternative partial sources to the data.

- The normalized mean bias is often used as a metric for model evaluation against observations. As pointed out by the referee, our notation was unclear so we have changed the equation to the following: "(NMB=($\overline{Model}/\overline{Obs}$ − 1)×100)".

Page 9, line 22-25. I assume that you are saying that, for the same size and place the lifetime is the same whatever the source (this must be true), but that the lifetime is longer because the blowing snow sourced aerosol tends to form when it's colder. It might be clearer if you explain it more in this way.

- Yes, the referee is correct. We have changed the wording in the revised manuscript to clarify this point:
"Open ocean SSA form over lower latitude warmer regions, while sea ice SSA emissions occur at higher latitudes under much colder conditions, with less efficient removal processes in mixed-phased and ice clouds. The current parameterization in GEOS-Chem assumes that in-cloud scavenging of SSA does not occur in cold clouds (T<258 K) (Wang et al., 2011), thus wintertime sea-ice generated SSA are only removed by below-cloud scavenging (which is slow for accumulation mode aerosols) and dry deposition."

Page 9, line 24. Just a question from my ignorance: wouldn't we expect sea salt to become an efficient ice nucleus below its eutectic (ie about 250K), when it would become a solid?

- The statement in the manuscript refers to what is currently assumed in GEOS-Chem. Recent laboratory studies have shown that sea salt can be act as an ice nuclei by deposition freezing or immersion freezing. This has been added to the revised manuscript:
"The current parameterization in GEOS-Chem assumes that in-cloud scavenging of SSA does not occur in cold clouds (T<258 K) (Wang et al., 2011), thus wintertime sea-ice generated SSA are only removed by below-cloud scavenging (which is slow for accumulation mode aerosols) and dry deposition. Recent laboratory studies have shown that SSA could act as ice nuclei by deposition freezing (Wise et al., 2012) and immersion freezing (DeMott et al., 2016), and might thus undergo in-cloud scavenging in mixed and ice clouds. This process is not currently included in GEOS-Chem. "

Fig 1 caption. Please explain here as well what NMB is: the reader should not have to read a quite difficult bit of text to understand the figure.

-We have made the change in the manuscript.

Fig 3 caption. I struggled to understand the text about the coloured circles. Do you mean that the larger circles (which are anyway hard to see) represent the ship's position between 15 and 19 April. If so, why not say this. If not, it needs a new explanation.

-The large circles indicate the locations of the ship between 15 and 19 April, and the color of the circle indicates the SSA mass concentrations observed on ICEALOT. We have added the clarification in the caption:
"The larger circles near Svalbard correspond to the location of the ship on 15-19 April, and they are color-coded based on observed SSA mass concentrations (same color scale are the model)."

Fig 4 and 5 caption, just for clarity please add "submicron" in the phrase "spatial distributions of wintertime submicron SSA".

-We have made the change in the manuscript.

Supplement, section 1. Much of this text duplicates what is already written on page 4, para 1.

-We have removed the duplicated text in the Supplement.

**Reference**

DeMott, P. J., Hill, T. C. J., McCluskey, C. S., Prather, K. A., Collins, D. B., Sullivan, R. C., Ruppel, M. J., Mason, R. H., Irish, V. E., Lee, T., Hwang, C. Y., Rhee, T. S., Snider, J. R., McMeeking, G. R., Dhaniyala, S., Lewis, E. R., Wentzell, J. J. B., Abbatt, J., Lee, C., Sultana, C. M., Ault, A. P., Axson, J. L., Martinez, M. D., Venero, I., Santos-Figueroa, G., Stokes, M. D., Deane, G. B., Mayol-Bracero, O. L., Grassian, V. H., Bertram, T. H., Bertram, A. K., Moffett, B. F., and Franc, G. D.: Sea spray aerosol as a unique source of ice nucleating particles, Proc. Natl. Acad. Sci., 113, 5797−5803, oi:10.1073/pnas.1514034112, 2016.

Huffman, G. J., Adler, R. F., Morrissey, M., Bolvin, D. T., Curtis, S., Joyce, R., McGavock, B., Susskind, J.: Global Precipitation at One-Degree Daily Resolution from Multi-Satellite Observations. J. Hydrometeorol., 2, 36–50, 2001.

Obbard, R. W., Roscoe, H.K., Wolff, E. W., and Atkinson, H. M.: Frost flower surface area and chemistry as a function of salinity and temperature, J. Geophys. Res., 114, D20305, doi:10.1029/2009JD012481, 2009.

Wang, Q., Jacob, D. J., Fisher, J. A., Mao, J., Leibensperger, E. M., Carouge, C. C., Le Sager, P., Kondo, Y., Jimenez, J. L., Cubison, M. J., and Doherty, S. J.: Sources of carbonaceous aerosols and deposited black carbon in the Arctic in winter-spring: implications for radiative forcing, Atmos. Chem. Phys., 11, 12453-12473, doi:10.5194/acp-11-12453-2011, 2011.

Wise, M. E., Baustian, K. J., Koop, T., Freedman, M. A., Jensen, E. J., and Tolbert, M. A.: Depositional ice nucleation onto crystalline hydrated NaCl particles: a new mechanism for ice formation in the troposphere, Atmos. Chem. Phys., 12, 1121-1134, doi:10.5194/acp-12-1121-2012, 2012.

Yang, X., Pyle, J. A., and Cox, R. A.: Sea salt aerosol production and bromine release: Role of snow on sea ice, Geophys. Res. Lett., 35, L16815, doi:10.1029/2008GL034536, 2008.

Yang, X., Pyle, J. A., Cox, R. A., Theys, N., and Van Roozendael, M.: Snow-sourced bromine and its implications for polar tropospheric ozone, Atmos. Chem. Phys., 10, 7763–7773, doi:10.5194/acp-10-7763-2010, 2010.